# On the Diminishing Reliability of Reference-Free Memorization Detection in Modern Diffusion Models

## Abstract

Diffusion models have been observed to memorize and regurgitate portions of their training data, which raises potential copyright and privacy concerns. To quantify and mitigate this phenomenon, various reference-free metrics that operate without training data access have become an effective tool for detecting memorization in text-to-image systems. As diffusion models expand beyond the familiar text-to-image paradigm to encompass multi-modal and multi-stage training for 3D and video synthesis, the reliability of existing detection methods in these novel domains remains unclear. In this work, (1) We find that metric efficacy declines when applied to models that are fine-tuned in multiple stages from a text-to-image base to support additional modalities, where more varied training protocols may obscure memorization signals from existing detection techniques. (2) We demonstrate that these metrics have limited reliability in distinguishing between successful and failed memorization mitigation attempts, risking false judgments in model sanitization efforts. (3) We trace this performance degradation to violations of assumptions underlying current detection frameworks and conduct factorized analysis. Our findings call for caution when applying existing memorization detection metrics beyond text-to-image models and point toward the need for more robust evaluation methods tailored to a wider range of emerging diffusion models with diverse training protocols.

## 1 Introduction

The remarkable success of diffusion models in generating high-quality images has been accompanied by growing concerns about their ability to unintentionally reproduce memorized training data (Carlini et al., 2023; Somepalli et al., 2023a; 2024). This could lead to the reproduction of copyrighted content, the leakage of sensitive information, and privacy violations of users who have contributed to the training data. These risks have motivated extensive research into memorization detection methods, with *reference-free* approaches attracting particular interest for their practicality, as they can operate without access to training data. Among these, methods leveraging the discrepancy between conditional and unconditional denoising trajectories of classifier-free guidance (CFG) (Ho & Salimans, 2022) have been embraced for their low computational cost and impressive efficacy (Wen et al., 2023; Jeon et al., 2024; Chen et al., 2025a; Ma et al., 2025).

Diffusion models have rapidly evolved beyond the standard text-to-image (T2I) applications for which these detection methods were developed and validated. More recent systems increasingly employ complex designs, including multi-stage training procedures (Shi et al., 2023; Yang et al., 2025), multi-modal loss objectives (Wang et al., 2023a), domain-specific regularization (Lin et al., 2025), and alternative parameterizations or schedulers (Karras et al., 2022). These methodological advances underpin diverse applications: video generation models use joint training on image and video data (Yan et al., 2024; Wang et al., 2023a), 3D-aware models incorporate geometric constraints or spatial priors (Kant et al., 2024; Li et al., 2023a), and safety-critical applications often demand post-hoc concept removal through unlearning (Gandikota et al., 2023; Kumari et al., 2023). Despite the transformation in the diffusion models research landscape, the evaluation of the effectiveness of memorization detection procedures is still largely focused on vanilla text-to-image models. It is

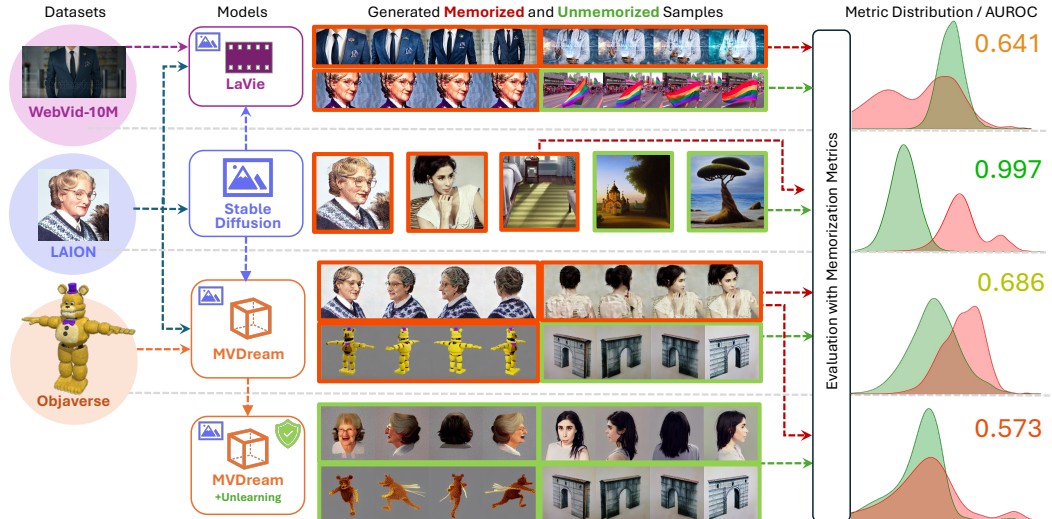

Figure 1: Memorization detection becomes more challenging in diffusion models with more varied training protocols. Detection metrics that work well on Stable Diffusion show decreased performance on its downstream models (i.e. weight initialization) like LaVie, MVDream, and models modified through unlearning techniques. This is reflected by the increased overlap between the distribution of the values of the metric used to distinguish memorized (red) and unmemorized (green) samples and thus, reduced detection AUROC scores.

therefore unclear whether these procedures remain effective across this wider variety of diffusion models, or if additional research efforts are required to produce more general detection procedures.

In this work, we systematically evaluate reference-free memorization detection methods across modern multi-modal diffusion models to address this critical gap. We demonstrate that complex training protocols can obscure memorization signals, potentially undermining the reliability of existing detection techniques. Our contributions are as follows:

1. **Widening metric efficacy evaluation.** We observe systematically reduced memorization detection efficacy across multi-modal DMs (Sec. 3.2), suggesting that complex training may blur memorization signals and affect detection reliability across different domains and training stages.

2. **Identifying unlearning evaluation challenges.** We find these metrics exhibit reduced reliability when distinguishing successful and failed sample removal attempts (Sec. 3.3), thus compromising the assessment of model sanitization efforts.

3. **Factorized analysis.** We link specific design choices of models' training protocols to potential violations of assumptions underlying existing memorization metrics (Sec. 4.4). Through measurements (Sec. 4.3) and controlled experiments (Sec. 5), we demonstrate how certain design choices common among modern multi-stage DMs could affect metric efficacy in predictable ways.

Our work reveals training complexity as a fundamental challenge for memorization detection. We conclude with recommendations for developing training-aware detection strategies and highlight the urgent need for robust verification methods in safety-critical applications.

## 2 RELATED WORK

### 2.1 DETECTING MEMORIZATION IN DIFFUSION MODELS.

Memorization phenomena in diffusion models have been extensively studied through extraction attacks (Carlini et al., 2023; Somepalli et al., 2023a), which directly reproduce training samples, and membership inference attacks (Wu et al., 2022; Hu & Pang, 2023; Pang et al., 2023), which reveal whether a given example was part of the training set. These findings confirm that text-to-image (T2I) diffusion models, and, specifically, Stable Diffusion (SD) (Rombach et al., 2022) variants can expose training content and thus pose copyright and privacy risks. Although prior work has begun exploring memorization in other modalities such as medical imaging (Rahman et al., 2024; Dar et al., 2023; 2024b; 2025; 2024a) and video synthesis (Chen et al., 2024b), systematic evaluation across the increasingly prevalent multi-stage, multi-modal training pipelines of modern DMs remains limited.

Among the various memorization detection strategies, reference-free approaches (Ren et al., 2024; Carlini et al., 2023; Hintersdorf et al., 2024; Ma et al., 2025; Brokman et al., 2025) are especially attractive because they do not require access to the original training data. In particular, CFG discrepancy-based methods (Wen et al., 2023; Jeon et al., 2024; Chen et al., 2025a), those that compare conditional and unconditional generation trajectories, have shown strong performance for T2I models. In Sec.3.2, we extend the evaluation of this class of metrics beyond the popular image domain, providing a more systematic assessment of their effectiveness on multi-modal DMs trained through multiple stages.

### 2.2 MITIGATING MEMORIZATION IN DIFFUSION MODELS.

In response to growing concerns related to the impact of training data extraction, interventions to mitigate memorization risks have been developed across various stages of the model development lifecycle. These approaches can be broadly categorized into pre-emptive and post-hoc strategies.

**Pre-emptive Mitigation.** Data-centric approaches that aim to limit the presence of replicated samples to reduce the likelihood of extraction, including semantic deduplication (Abbas et al., 2023) and curation of datasets complying with copyright law (Gokaslan et al., 2023). Training-time strategies include compositionally isolated training (Golatkar et al., 2023), despecification guidance (Chen et al., 2024a), and replication-aware architectures (Li et al., 2024b).

**Post-Training Mitigation.** Post-training approaches offer practical advantages: they apply to deployed models, require no training data access, and enable targeted content removal without retraining. These methods include concept ablation (Kumari et al., 2023), gradient-based erasure (Zhang et al., 2023; Wu et al., 2024a), scalable batch removal (Fan et al., 2023; Lu et al., 2024), attention reweighting (Ren et al., 2024), token masking (Chen et al., 2025a), neuron suppression (Hintersdorf et al., 2024), and guided sampling (Dong et al., 2023).

Given their utility for model sanitization, ease of adoption, and flexibility, post-training methods represent powerful tools for ensuring model safety. However, realizing this potential requires accurate evaluation to confirm that memorized content has actually been removed. This challenge motivates our experiments in Sec. 3.3, which evaluate whether reference-free metrics can reliably assess the effectiveness of memorization mitigation techniques.

## 3 EVALUATION

To complement prior work in Sec. 2.2, which has largely focused on T2I models and to address the open question of whether those findings generalize to a wider range of diffusion models with more diverse training protocols, we conduct a two-part evaluation: (1) In Sec. 3.2, we assess the performance of five prominent reference-free memorization metrics across three naturally multi-stage trained 3D and video diffusion models that deviate from the standard T2I LDM paradigm. (2) In Sec. 3.3, we evaluate the capability of these metrics to reliably evaluate twelve post-training memorization mitigation techniques reviewed in Sec. 2.2 and Apx. B.3.

We begin by describing how we collected sets of high-risk training samples that are likely to be memorized across three datasets in Sec. 3.1.1 (further details in Apx. A.2), and by briefly reviewing three families of reference-free memorization metrics (further details in Apx. B.2).

### 3.1 PRELIMINARIES: DATA AND METRICS

#### 3.1.1 HIGH-RISK TRAINING SAMPLE COLLECTION.

We consider two types of memorization described by Webster et al. (2023) and widely adopted in the memorization literature (Wen et al., 2023; Jeon et al., 2024). *Verbatim Memorization* refers to the model reproducing training samples with minimal or no variation, often at the pixel level or under only trivial transformations such as resizing or compression. *Template Memorization* refers to the model reproducing a structural or compositional layout from the training data while varying superficial local details such as textures, colors, or minor object attributes. In our setting, for video and 3D DMs, the distinction between these categories becomes blurred: in WebVid10M (Bain et al., 2021), large numbers of stock videos share the same motion or scene template with only small

differences, while in Objaverse (Deitke et al., 2023), identical mesh geometries are frequently paired with different textures. Given this prevalence of ambiguous cases, we treat both forms jointly.

For LAION, we use the established benchmark from Webster (2023), consisting of 500 prompts known to elicit memorized images and 500 non-memorizing prompts, though we relabel samples since only a subset of memorization can be retained and transferred to downstream models. For newly acquired memorization from domain-specific fine-tuning on Objaverse and WebVid-10M, we establish a reproducible pipeline to identify samples with many **near-duplicates** in language and/or visual modalities, making them prone to memorization. For each high-risk prompt identified, we perform inference across four random seeds. The resulting generated outputs then go through a semi-automated labelling process to establish a reliable ground truth for evaluation. The complete pipeline for both high-risk sample collection and labelling is detailed in Appendix A.2.

### 3.1.2 REFERENCE-FREE MEMORIZATION METRICS.

We evaluate the AUROC of nine metrics spanning three categories, each exploiting different signals that distinguish memorized samples from their non-memorized counterparts. Complete definitions and implementation details are provided in Appendix B.2.

We begin with score-based metics for memorization detection. These methods explicitly or implicitly measure the sharpness, curvature, or shape of the learned probability landscape to detect memorization. In particular, the first four metrics in this group share a common mechanism: they quantify the discrepancy between conditional and unconditional diffusion trajectories, positing that memorized samples exhibit stronger prompt dominance over noise influence. The latter two metrics assess the intrinsic properties of the sample's location on the manifold without relying on an unconditional baseline, yet remain closely related as they probe the same underlying geometric structure.

- **Noise Difference Norm (NDN)** (Wen et al., 2023): measures the magnitude of difference between conditional and unconditional noise predictions $D = \frac{1}{T} \sum_{t=1}^{T} \left\| \epsilon_\theta(x_t, t, e_p) - \epsilon_\theta(x_t, t, e_\emptyset) \right\|_2$, where $x_t$ is the noisy latent at timestep $t$, $e_p$ is the conditional text embedding for prompt $p$, and $e_\emptyset$ is the unconditional (empty) text embedding.

- **Hessian Eigenvalue Difference (HED)** (Jeon et al., 2024): approximates differences in Hessian eigenvalue magnitudes between conditional and unconditional score functions using directional finite differences. Large magnitude differences indicate regions where conditioning creates sharp, localized probability peaks, which are characteristic of memorized content. We measure them at three DDIM timesteps ($t_{\text{DDIM}} \in \{50, 20, 1\}$).

- **Bright Ending (BE)** (Chen et al., 2025a): a spatially-aware variant of NDN that uses cross-attention weights to the final prompt token as a localization mask, computing the attention-weighted noise difference norm to identify regions where memorization may be occurring.

- **SSIM of Noise Differences** (Hintersdorf et al., 2024): computes the Structural Similarity Index (SSIM) Wang et al. (2004) between noise differences across random seeds, where higher SSIM values indicate more consistent denoising trajectories and hence, stronger memorization.

- **InvMM** (Ma et al., 2025): measures memorization by inverting a sensitive latent noise distribution that can replicate the target image, quantified as the minimum KL divergence between this inverted distribution and the standard Gaussian prior. While distinct from score-based, it is fundamentally coupled to the score function's geometry.

- **pLaplace** (Brokman et al., 2025): employs the $p$-Laplacian operator to measure the intrinsic curvature of the learned probability landscape, similarly to HED (Jeon et al., 2024).

We next evaluate diversity-based metrics, which measure the semantic or visual consistency across multiple generations from the same prompt, where low diversity indicates memorization:

- **Median SSCD** (Hintersdorf et al., 2024): computes median cosine similarity between SSCD embeddings across multiple generations from the same prompt.

- **Tiled $\ell_2$ (TL2)** (Carlini et al., 2023): evaluates minimum pairwise distance between tiled image patches across different seeds.

- **SSIM of Noise Differences** (Hintersdorf et al., 2024) also falls into this category.

Finally, we include **attention-based** metrics that analyse how memorization impact cross-attention pattern characteristics:

- **Cross-Attention Entropy (CAE)** (Ren et al., 2024): quantifies the dispersion of cross-attention weights to identify memorization with two variants: (i) *Global CAE-D* and (ii) *Layer-wise CAE-E*. This metric builds on the observation that memorized samples maintain concentrated attention on trigger tokens while while non-memorization progressively shifts attention towards the beginning token, serving as a robust indicator of memorization.

- **Bright Ending (BE)** (Chen et al., 2025a) can also be considered a member of this category.

### 3.2 METRIC EFFICACY ON NATURALLY TRAINED MODELS

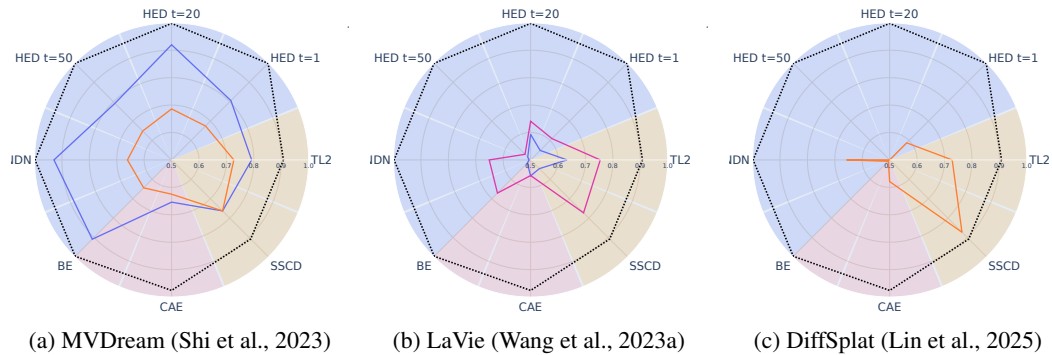

(a) MVDream (Shi et al., 2023)    (b) LaVie (Wang et al., 2023a)    (c) DiffSplat (Lin et al., 2025)

Figure 2: Metric AUROC on naturally trained multi-stage DMs. Each radial axis corresponds to a memorization metric defined in Appendix B.2. Higher values (larger radii) indicate better separation between memorized and non-memorized samples. Solid coloured lines indicate AUROC on LAION, Objaverse and WebVid-10M, which exhibit notable degradation compared to the SD 1.4 baseline (dotted grey lines).

**Setup.** In our first set of experiments, we evaluate the efficacy of reference-free metrics across three diffusion models: MVDream (Shi et al., 2023), LaVie (Wang et al., 2023a), and DiffSplat (Lin et al., 2025). All three have a SD backbone, meaning they may inherit memorized samples from the original LAION training data, while also potentially acquiring new memorized samples from their respective domain-specific fine-tuning datasets. Their training protocols involve multi-stage learning, joint-modality training, or domain-specific regularizations (more details in Appendix A.1.1).

**Results.** Fig. 2 shows consistent metric AUROC degradation across all three models compared to the standard text-to-image (SD 1.4) baseline (gray dotted line). Most metrics exhibit substantially reduced ability to distinguish memorized from non-memorized content, suggesting that complex training protocols can systematically obscure the signals these metrics rely on. For models with significant memorization from both pre-training and fine-tuning stages (MVDream and LaVie), metric performance varies inconsistently across the two data sources. This suggests that memorization signals may differ depending on when and how the content was learned during training. In contrast, DiffSplat shows no significant transferred memorization from the LAION pre-training stage. Two factors likely contribute to this observation: (1) the modality gap between images and native 3D Gaussian Splats is substantially larger than that between images and multi-view images or multi-frame videos, and (2) strong domain-specific regularization is applied during fine-tuning. This suggests that fine-tuning with task-specific regularizers that diverge from the original training objective may unintentionally mitigate inherited memorization.

### 3.3 METRIC EFFICACY FOR MITIGATION EVALUATION.

**Setup.** An important application for these metrics is the evaluation of memorization mitigation strategies, where a metric is required to reliably confirm whether a sample has been successfully erased from a model. We test this capability by applying twelve different post-hoc memorization mitigation methods (see Table 6 for a summary and Appendix B.3 for a brief review) to MVDream (Shi et al., 2023), on both inherit memorized samples from LAION and newly acquired memorized samples from Objaverse. We then use the same set of metrics to assess the outcome and

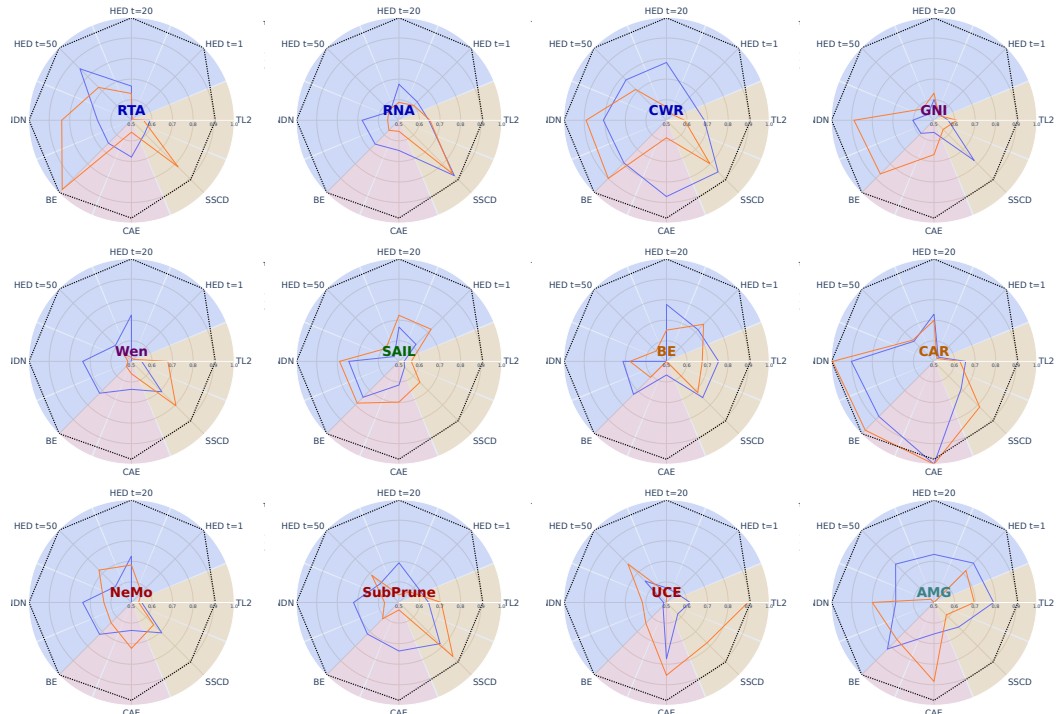

Figure 3: Metric AUROC for memorization mitigation evaluation. Each radial plot corresponds to a different mitigation method in Table 6, where font colours indicate method categorization. Memorization metrics are marked around the periphery. Coloured lines indicate inherited memorized samples from LAION pre-training and newly acquired memorized samples from Objaverse with the gray dotted circle marking the SD 1.4 baseline performance. Higher values indicate stronger ability to distinguish between classes. The widespread shrinkage from baseline shows that most metrics cannot reliably evaluate memorization removal.

plot their AUROC, assessing their capability of differentiating successfully unlearned concepts from original memorized samples and unsuccessful attempts where memorization persists.

**Results.** As shown in Figure 3, the results reveal concerning limitations in the evaluation of memorization mitigation methods: reference-free metrics often fail to reliably distinguish between successful and unsuccessful concept removal attempts. In many cases, their discriminative ability is even worse than on naturally trained models, suggesting that certain mitigation methods may further obscure detection signals. A few exceptions occur in "white-box" settings, where the metric itself (typically via attention manipulation) is explicitly targeted as part of the mitigation objective. These findings have important implications, as unreliable evaluation can lead to false judgments about a model's memorization behaviours and the efficacy of sanitization efforts.

## 4 DISCUSSIONS

The previous sections demonstrate that the efficacy of reference-free memorization metrics declines on models with varied training protocols. To provide a principled explanation for this phenomenon, we leverage the geometric frameworks pioneered by Jeon et al. (2024); Kamkari et al. (2024); Ross et al. (2025); Buchanan et al. (2025). Our analysis examines how novel training designs cause models to diverge from the framework's core assumptions, in turn explaining the observed drop in detection accuracy. For the remainder of largely empirical metrics not covered by this framework, we provide a speculative discussion in Appendix B.2.

Our analysis proceeds as follows: in Sec. 4.1, we review the geometric framework by Jeon et al. (2024) and identify its key theoretical assumptions in Sec. 4.2. We then empirically measure the degree to which these assumptions are met on production-scale models in Sec. 4.3. Finally, in Sec. 4.4, we connect the observed violations to specific training design choices.

## 4.1 GEOMETRIC FRAMEWORK FOR MEMORIZATION

We briefly revise the framework developed by Jeon et al. (2024), which identifies geometric properties of the model's learned probability landscape as a prominent memorization indicators. Let $D_t^2 = \|\epsilon_\theta(x_t, e_p) - \epsilon_\theta(x_t, e_\emptyset)\|_2^2$ denote NDN at timestep $t$, where $\epsilon_\theta(x_t, e_p)$ and $\epsilon_\theta(x_t, e_\emptyset)$ are the conditional and unconditional noise predictions for a noisy latent $x_t$ corrupted with noise $\epsilon_t$.

From a geometric perspective, these noise predictions correspond to *score functions*. Here $p_t(\cdot|c)$ and $p_t(\cdot)$ denote the conditional and unconditional distributions of $x_t$ at timestep $t$.

$$s_\theta(x_t, e_p) = -\frac{\epsilon_\theta(x_t, e_p)}{\sqrt{1-\bar{\alpha}_t}} = \nabla_{x_t} \log p_t(x_t|c); \quad s_\theta(x_t, e_\emptyset) = -\frac{\epsilon_\theta(x_t, e_\emptyset)}{\sqrt{1-\bar{\alpha}_t}} = \nabla_{x_t} \log p_t(x_t). \quad (1)$$

Their Hessians capture local curvature:

$$H_c(x_t) = \nabla_{x_t}^2 \log p_t(x_t|c); \qquad H_u(x_t) = \nabla_{x_t}^2 \log p_t(x_t). \quad (2)$$

We also define the *local covariance matrices* of the conditional and unconditional distributions at timestep $t$ as

$$\Sigma_{t,c} = \mathrm{Cov}_{p_t(\cdot|c)}[x_t], \quad \mu_c = \mathbb{E}_{p_t(\cdot|c)}[x_t]; \qquad \Sigma_t = \mathrm{Cov}_{p_t(\cdot)}[x_t], \quad \mu = \mathbb{E}_{p_t(\cdot)}[x_t]. \quad (3)$$

where $\mu_c$ and $\mu$ are the corresponding means.

**Memorization as Geometric Sharpness.** Jeon et al. (2024) theorize that memorized samples correspond to sharp, isolated peaks in the probability landscape, characterized by: (i) large negative eigenvalues ($\lambda_{\mathrm{mem}} \ll \lambda_{\mathrm{unmem}} < 0$), (ii) high curvature concentration ($\mathrm{tr}(H_c^2) \gg \mathrm{tr}(H_u^2)$), and (iii) separable eigenvalue distributions between memorized and unmemorized samples.

**Unifying Sharpness, NDN and BE Metrics.** The squared score difference can be expressed as:

$$\mathbb{E}[D_t^2] = \mathbb{E}[\|s_\theta(x_t, e_p) - s_\theta(x_t, e_\emptyset)\|^2] \propto \mathbb{E}[\mathrm{tr}((H_c - H_u)^2)] \approx \sum_{i=1}^{d} \frac{(\lambda_{i,c} - \lambda_{i,u})^2}{\lambda_{i,c}} \quad (4)$$

which, under a local Gaussian approximation, reduces to the normalized differences of eigenvalues where $\lambda_{i,c}$ and $\lambda_{i,u}$ are eigenvalues of $H_c$ and $H_u$, and $d$ stands for the dimensionality of the latent space. Since $\epsilon_\theta$ and $s_\theta$ are linearly related, NDN (Wen et al., 2023) and localized variants like BE (Chen et al., 2025a) similarly reflect the underlying differential curvature as Hessian-based sharpness metrics proposed by Jeon et al. (2024).

## 4.2 CORE ASSUMPTIONS FOR RELIABLE DETECTION

The geometric framework described in Sec. 4.1 (and by extension, the CFG-based metrics built upon it) relies on a set of specific structural assumptions about the model's learned probability landscape. When modern diffusion model training deviates from these conditions, the framework may lose its explanatory power, undermining the reliability of memorization detection methods built upon it. Table 1 summarizes these key assumptions, which we test empirically in the following sections. We further discuss how these assumptions support the framework of Jeon et al. (2024) in Apx. **??**.

Table 1: Core assumptions underlying CFG discrepancy-based memorization detection methods.

| Assumption | Formulation | Interpretation |
|---|---|---|
| (A1) Unbiased Score Estimation | $\mathbb{E}[s_\theta(x_t, c) - \nabla_{x_t} \log p_t(x_t|c)] = 0$ | Score functions accurately reflect probability gradients |
| (A2) Gaussian Local Structure | $\mathbb{E}[\|s_\theta(x_t, c)\|^2] = -\mathrm{tr}(H_c(x_t))$ | Local probability distributions approximate Gaussian |
| (A3) Sharpness Persistence | $\{\lambda_i\}_{T-1}$ patterns correlate with later timesteps | Memorization signals persist through reverse process |
| (A4) Covariance Commutativity | $\Sigma_t \Sigma_{t,c} = \Sigma_{t,c} \Sigma_t$ and $\mu = \mu_c$ | Cond and uncond covariances have aligned eigenspaces |
| (A5) Mean-Field Gaussian Prior | $x_T \sim \mathcal{N}(0, I)$ | Unstructured, isotropic starting point for generation |
| (A6) Boundary Regularity | $\lim_{\|x\| \to \infty} p(x)s(x) = 0; \mathbb{E}[\|s(x)\|^2] < \infty$ | Score func vanish at infinity with finite second moments |

## 4.3 EMPIRICAL VALIDATION VIA ASSUMPTION DIAGNOSTIC MEASUREMENTS

Having established the core assumptions of the geometric framework in Sec. 4.2, we now empirically investigate whether they hold in practice for the models evaluated in Sec. 3. By measuring the degree to which each model satisfies these assumptions, we can explore the relationship between

assumption compliance and the observed efficacy of memorization detection metrics. We apply diagnostic measurements (Table 2) to all models evaluated in Section 3 and report results in Table 3. Detailed definitions and implementation notes for each diagnostic are provided in Appendix C.2.

Table 2: Core geometric assumptions and their corresponding measurable proxies. The rightmost column indicates whether higher ($\uparrow$) or lower ($\downarrow$) values reflect stronger adherence to the assumption.

| Assumption | Diagnostic Measurements | Method | $\Delta$ |
|---|---|---|---|
| (A1) Unbiased Score Estimation | Score Matching Consistency | $(1 + \lvert\nabla \cdot s + 0.5\lvert s\rvert^2\rvert/\lvert s\rvert)^{-1}$ with Hutchinson's estimator. | $\uparrow$ |
| (A2) Gaussian Local Structure | Score-Curvature Pearson Corr. | Pearson's $r$ between $\lVert s_\theta(x_t, c)\rVert^2$ and $-\mathrm{tr}(H_c(x_t))$. | $\uparrow$ |
| (A3) Sharpness Persistence | Temporal Autocorrelation | Temporal autocorrelation of Hessian magnitudes across timesteps. | $\uparrow$ |
| (A4) Covariance Commutativity | Eigenspace Alignment | Mean singular value of the inner product of $\mathbf{V}_c^T \mathbf{V}_u$. | $\uparrow$ |
| (A5) Mean-Field Gaussian Prior | Gaussian Prior $p$-value | Kolmogorov-Smirnov test of initial latents against $\mathcal{N}(0, I)$. | $\uparrow$ |
| (A6) Boundary Regularity | Score Explosion Indicator | Ratio of max to mean noise prediction magnitude. | $\downarrow$ |

Table 3: Assumption diagnostic measurements across diffusion models. Each row evaluates how well an assumption (A1)-(A6) is satisfied using its corresponding diagnostic measurement. Columns represent different models. The final row shows HED AUROC scores. Green shading indicates better assumption adherence.

| Assumption | SD 1.4 | LaVie | SD 1.5 | MVDream | DiffSplat |
|---|---|---|---|---|---|
| (A1) Unbiased Score Estimation | $0.496 \pm 0.004$ | $0.319 \pm 0.044$ | $0.497 \pm 0.004$ | $0.411 \pm 0.007$ | $0.447 \pm 0.007$ |
| (A2) Gaussian Local Structure | $0.940 \pm 0.037$ | $0.748 \pm 0.017$ | $0.940 \pm 0.037$ | $0.403 \pm 0.017$ | $-0.820 \pm 0.128$ |
| (A3) Sharpness Persistence | $0.374 \pm 0.055$ | $0.120 \pm 0.015$ | $0.374 \pm 0.055$ | $0.240 \pm 0.0403$ | $0.339 \pm 0.108$ |
| (A4) Covariance Commutativity | $0.819 \pm 0.028$ | $0.626 \pm 0.054$ | $0.816 \pm 0.033$ | $0.763 \pm 0.050$ | $0.436 \pm 0.014$ |
| (A5) Mean-Field Gaussian Prior | $0.812 \pm 0.215$ | $0.800 \pm 0.263$ | $0.812 \pm 0.215$ | $0.796 \pm 0.247$ | $0.774 \pm 0.183$ |
| (A6) Boundary Regularity | $0.011 \pm 0.003$ | $0.025 \pm 0.002$ | $0.012 \pm 0.002$ | $0.006 \pm 0.002$ | $0.002 \pm 0.001$ |
| **HED AUROC** | **0.998** | **0.641** | **0.997** | **0.686** | **0.588** |

Table 3 presents the diagnostic measurements (mean $\pm$ standard deviation) for each assumption (rows) across the different models (columns). The final row reports HED metric AUROC scores, enabling a direct comparison between the degree of assumption satisfaction and detection efficacy. For the baseline SD model variants, the diagnostics show generally good adherence to the theoretical assumptions. This provides evidence for the validity of the geometric framework on standard text-to-image models. In contrast, models with more varied training protocols exhibit more noticeable deviations. LaVie, for instance, shows systematically weaker adherence to assumptions (A1)-(A4), and this degradation is associated with a substantial drop in its AUROC score. MVDream shows a similar, though less pronounced, pattern of assumption violation and performance decline. This suggests that the training procedures used for these multi-modal models may alter the underlying geometry of the probability landscape in ways that violate the framework's prerequisite conditions.

### 4.4 POTENTIAL IMPACT OF DESIGN CHOICES ON METRIC EFFICACY

Building on the finding that varied training protocols correlate with assumption violations and reduced detection performance, we conduct a factorized analysis. We identify four common design choices in modern diffusion models and map them to the potential deviations from the core assumptions outlined in Table 1. Each design choice, while advancing generative capabilities, can inadvertently violate the assumptions that CFG discrepancy-based metrics rely upon.

Table 4: A non-exhaustive mapping between design choices (columns) and potential assumption violations (rows). Each $\times$ indicates that at least one implementation of a design choice can violate the assumption.

| Assumption | (D1) Non-Standard Objectives | (D2) Multi-Stage Distribution Shift | (D3) Non-Monotonic Schedulers | (D4) Structured or Biased Priors |
|---|---|---|---|---|
| (A1) Unbiased Score Estimation | $\times$ | | | $\times$ |
| (A2) Gaussian Local Structure | $\times$ | $\times$ | $\times$ | $\times$ |
| (A3) Sharpness Persistence | | $\times$ | $\times$ | |
| (A4) Covariance Commutativity | | $\times$ | $\times$ | $\times$ |
| (A5) Mean-Field Gaussian Prior | | | | $\times$ |
| (A6) Boundary Regularity | Generally not violated by reasonably trained and converged model. | | | |

**(D1) Non-Standard Objectives.** Adding auxiliary losses, such as the geometric regularizers of DiffSplat (Lin et al., 2025), forces the model to optimize beyond pure score matching. As the model

is no longer estimating solely the data log-probability, score estimates may become biased (A1). This can also distort local geometry (A2) by imposing structures not implied by the data likelihood.

**(D2) Multi-Stage Distribution Shift.** Fine-tuning a model on a new domain (e.g., images → videos for LaVie Wang et al. (2023a) and images → 3D for MVDream Shi et al. (2023)) alters the inherited probability landscape in ways that need not be uniform. Such shifts can weaken local Gaussianity (A2) and disturb the alignment of conditional and unconditional curvature directions, thereby affecting sharpness persistence (A3) and covariance eigenspace alignment (A4). These effects may depend on the extent and stage of the shift, suggesting that violations of geometric assumptions can emerge in complex and potentially non-monotonic patterns as a function of intervention strength.

**(D3) Non-Monotonic Schedulers.** Novel schedulers (Karras et al., 2022) that induce non-monotonic noise levels can alter the trajectory of the reverse process. Such changes may destabilize memorization signals across timesteps, with possible effects on sharpness persistence (A3), eigenspace alignment (A4), and local Gaussian structure (A2).

**(D4) Structured or Biased Priors.** Non-isotropic initializations directly violate (A5) but can also interact with other assumptions. Priors with *spatial structures* (e.g., 3D object-centricity (Kant et al., 2024; Li et al., 2023a)) or *frequency structures* (Rahaman et al., 2019)) may introduce distinct geometric fingerprints, the nature of which we examine in controlled settings (Fig. 4).

## 5 CONTROLLED EXPERIMENTS

While the empirical analysis in Sections 3–4 demonstrates potential effects of complex training protocols on metric reliability, real models often incorporate multiple design choices simultaneously, making it difficult to isolate individual contributions. To better understand these effects, we conduct controlled experiments targeting the four design choices (D1)-(D4) identified in Section 4.4. We investigate (1) how each design choice affects model adherence to geometric assumptions, and (2) how assumption violations correlate with detection performance degradation.

**Experimental Setup.** All experiments train diffusion models using CIFAR-10 (Krizhevsky et al., 2009) with identical UNet architectures. We create memorized samples by duplicating randomly selected images 100 times each with instance-specific prompts ("a high-resolution photograph of CIFAR image #ID"), alongside unmemorized images with generic class prompts ("a photograph of a CLASS"). This protocol follows established observations (Carlini et al., 2023; Somepalli et al., 2023a) that training data duplication and highly specific prompts promote memorization. Each configuration is repeated across 5 random seeds.

**Evaluation Protocol.** For each model, we quantify adherence to geometric assumptions using proxy tests from Table 2 and metric efficacy using AUROC of the HED metric (Jeon et al., 2024), which directly embodies the geometric framework and shares similar trends as other CFG-based metrics.

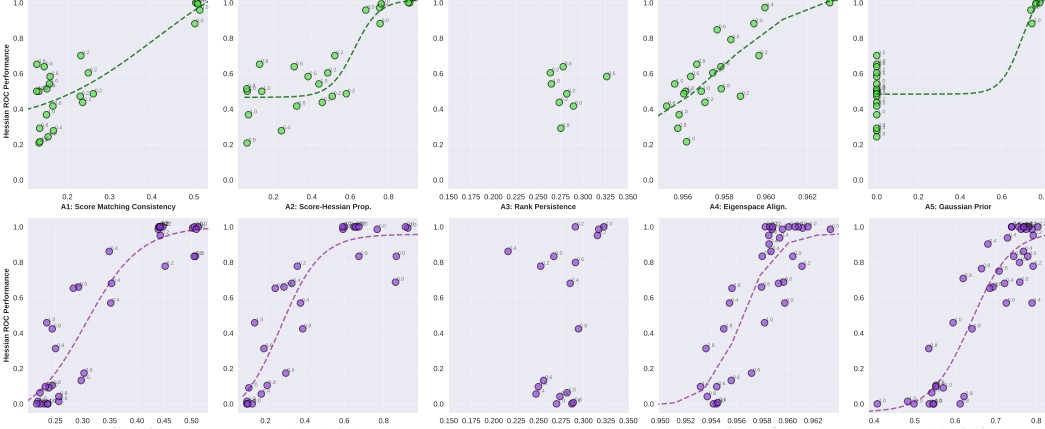

Figure 4: Detection performance under interventions targeting (D4) structured prior. Top: spatial radial-decay priors. Bottom: frequency-biased priors. Each point represents a trained model, plotting Hessian metric AUROC (y-axis) against adherence to a specific geometric assumption (x-axis). Relationships between metric performance and assumption adherence mirror Table 4, Column (D4).

**Results and Analysis.** Our controlled experiments, which we focus here on Structured Priors (D4) for brevity (see Appendix D for D1-D3), reveal how specific design choices could degrade detection performance by violating a subset of assumptions. Both prior types exhibit similar high-level patterns of impact: they strongly affect adherence to (A1), (A2), and (A5), while having a comparatively minimal impact on (A3). On the contrary, their (A4) behaviours diverge. The correlation between metric performance and A4 adherence is much stronger for frequency-biased priors than for spatial priors. **Spatial biases** (top row) cause a milder degradation in AUROC, with the worst-performing models still achieving scores above 0.2. In contrast, **frequency biases** (bottom row) induce a far more brittle and catastrophic collapse, with many models failing completely.

Beyond these (D4)-specific observations, the results also highlight two broader patterns shared across all interventions. First, the **assumptions are interdependent**, with a single design choice often causing a cascade of violations. Second, the relationships between adherence and performance are often strongly **non-linear** and exhibit a sharp threshold effect with a "step-function"-like relationship. These qualitative trends support our central hypothesis, though we note that inherent noise in our proxy measurements makes precise cross-intervention quantitative comparisons challenging.

## 6 CONCLUSIONS

In this work, we have extended the evaluation of reference-free memorization detection methods to modern multi-modal diffusion models and find consistent degradation in their efficacy (Sec. 3.2) and that they cannot serve as reliable evaluation tools for memorization mitigation efforts (Sec. 3.3). To understand this phenomenon, we link metric degradation to violations of assumptions underlying existing detection frameworks (Sec. 4.4). We validate these insights through observational measurements (Sec. 4.3) on large-scale DMs as well as controlled experiments (Sec. 5), demonstrating how different training protocol modifications violate assumptions underlying theoretical efficacy guarantees and thus affect detection performance in predictable ways. Future work might explore training-aware detection strategies or develop alternative theoretical frameworks tailored for complex training regimes. Our work represents an initial step toward understanding how memorization detection scales to increasingly sophisticated generative models. We hope these insights contribute to ongoing efforts to develop reliable safety tools for emerging diffusion architectures.

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

**Large Language Models Usage Statement.** LLMs were used to polish writing and format tables.

# A ADDITIONAL NOTES ON EVALUATED MULTI-STAGE DMS

## A.1 TRAINING AND INFERENCE DETAILS OF EVALUATED MULTI-STAGE DMS

### A.1.1 TRAINING PROTOCOLS

MVDream (Shi et al., 2023)employs a multi-modal training strategy that fine-tunes SD 1.5 and 2.1 on both 2D and 3D data. The model trains jointly on the Objaverse dataset (800K 3D objects, each with 4 orthogonal rendered views) and a subset of LAION images. During second-stage training, LAION data is sampled with 30% probability and treated as standard 2D text-to-image generation by disabling 3D attention mechanisms and camera embeddings. For 3D data, the original 2D self-attention layers are inflated into 3D cross-view attention layers with inherited weights, while camera parameters are embedded via a 2-layer MLP and injected into time embeddings. This creates an implicit task weighting of approximately 30% image generation and 70% multi-view 3D generation through the sampling strategy.

LaVie (Wang et al., 2023a) implements joint image-video training to prevent catastrophic forgetting when extending SD 1.4 backbone to video generation. The model trains on WebVid10M for video data and LAION for image data using an explicit weighted objective:

$$\mathcal{L} = \mathbb{E}[\|\epsilon - \epsilon_\theta(\xi(v_t), t, c_v)\|^2] + \alpha \mathbb{E}[\|\epsilon - \epsilon_\theta(\xi(x_t), t, c_I)\|^2]$$

The first term represents video loss and the second represents image loss with weighting factor $\alpha$. Images are concatenated along the temporal axis to form multi-frame sequences, but temporal attention mechanisms are disabled for image portions of the batch. The architecture incorporates temporal self-attention layers with rotary positional encoding to capture video dynamics while preserving 2D generation capabilities.

These two models deviate from standard single-task LDM training through weighted multi-modal objectives that combine different data modalities (2D/3D for MVDream, image/video for LaVie) within the same training process, creating complex conditioning environments that stary away from the ideal environment for CFG discrepancy-based memorization metrics.

In contrast, DiffSplat (Lin et al., 2025) uses explicit domain-specific regularization during second-stage training. Built on SD v1.5, DiffSplat generates 3D Gaussian splats by fine-tuning the pre-trained T2I model with a composite objective:

$$\mathcal{L}_{\text{DiffSplat}} := \lambda_{\text{diff}} \cdot \mathcal{L}_{\text{diff}} + \lambda_{\text{render}} \cdot w_r(t) \cdot \mathcal{L}_{\text{render}}(D_{\phi_d}(F_\psi(\tilde{z}, t)))$$

where $\mathcal{L}_{\text{diff}}$ is the standard diffusion loss on splat latents (Gaussian splat properties encoded into the VAE latent space) and

$$\mathcal{L}_{\text{render}}(\mathcal{G}) := \frac{1}{V} \sum_{v=1}^{V} \left( \mathcal{L}_{\text{MSE}}(I_v, I_v^{\text{GT}}) + \lambda_p \cdot \mathcal{L}_{\text{LPIPS}}(I_v, I_v^{\text{GT}}) + \lambda_\alpha \cdot \mathcal{L}_{\text{MSE}}(M_v, M_v^{\text{GT}}) \right)$$

is a 3D rendering loss designed to enforce geometric consistency across arbitrary viewpoints. This rendering loss includes perceptual loss components and mask losses to reduce translucent artifacts, regularizing the predictor away from pure noise prediction toward 3D-coherent splat generation.

Table 5: Default Inference Hyperparameters of Evaluated DMs.

| Hyperparameter | MVDream | LaVie | DiffSplat |
|---|---|---|---|
| Scheduler | DDIM | DDIM | DDIM |
| Inference Steps | 50 | 50 | 20 |
| Guidance Scale | 7.5 | 7.5 | 7.5 |
| View/Frames | 4 | 16 | 4 |

### A.1.2 A BRIEF REMARK ON MEMORIZATION BIAS

Memorization bias refers to the tendency and extent to which a diffusion model reproduces training data rather than generating novel content. Unlike simple binary classification (memorized vs.

not memorized), we observe that memorization exists along multiple dimensions that significantly impact detection accuracy.

We identify three relatively orthogonal dimensions for characterizing memorization bias, ordered by increasing resolution from finest to most coarse:

**Aspect 1: Sample-Level Memorization Intensity.** Following Webster et al. (Webster et al., 2023) we have qualitatively defined *Verbatim Memorization* and *Template Memorization* Sec. 3. Template memorization represents a transition between memorization and generalization with both types of behaviours present and tends to be more changeling to detect and represents a less "intense" form of memorization (**?**Chen et al., 2025a).

**Aspect 2: Prompt-Level Memorization Consistency.** This measures how consistently a model memorizes the same content across different random seeds for a given prompt. Prompt with most seeds producing memorized outputs represent strong, consistent memorization that can be triggered by a variety of seeds. In contrast, sporadic memorization that is only tirggered by specific prompt-seed combinations is considered weaker and more challenging to detect. Metrics that rely on cross-seed diversity (e.g., LPIPS between different seeds) are more likely to fail when memorization is inconsistent, as the model occasionally "escapes" to generate diverse, non-memorized outputs.

**Aspect 3: Dataset-Level Memorization Prevalence.** The overall proportion of memorized samples in the training dataset creates a class imbalance problem for detection. This prevalence is influenced by data (repetition, presence of outliers, overly specific prompts, or templated content) and model (over-parameterization, insufficient regularization, or architectural choices that increase overfitting susceptibility) characteristics.

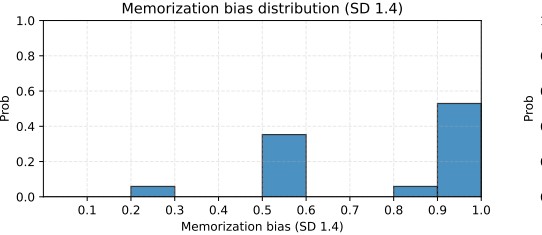 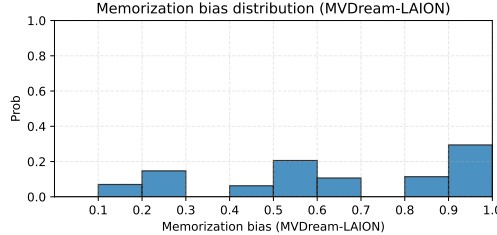

Figure 5: Memorization bias distributions comparing single-stage (SD 1.4, left) and multi-stage (MVDream-LAION, right) diffusion models. Memorization bias measures the proportion of seeds that trigger memorization for a given prompt. For inherited memorization from pre-trained datasets, there is a slight reduction in memorization bias. For newly acquired memorization from the second-stage dataset, memorization bias remains comparable to single-stage models.

**Observation 1: Comparable Per-Prompt Memorization Rates (Aspect 1).** For inherited memorization from pre-trained datasets (e.g., LAION), there is a slight reduction in memorization bias. The second-stage training appears to weaken, though not eliminate, the memorization tendencies established during pre-training. For newly acquired memorization from the second-stage dataset (e.g., Objaverse), memorization bias remains comparable to single-stage models. When a prompt triggers memorization, it typically does so consistently across seeds in both cases. The more pronounced differences between model types emerge along Aspect 2 (memorization intensity), which we discuss next.

**Observation 2: The Memorization-Generalization Continuum (Aspect 2).** In multi-stage models for video and 3D generation, the boundary between memorization and generalization becomes blurred. We observe a continuum where memorized content exhibits generalization-like behaviors, making detection particularly challenging. This continuum manifests through multiple mechanisms, including but not limited to:

*(1) Inherited Memorization with Multi-Modal Transformation.* When models like MVDream and LaVie fine-tune pretrained 2D models, they inherit memorized content from the base model but transform it through the new modality:

- **Many-to-one memorization**: Multiple memorized 2D images are merged, interpolated, or blended into a single multi-view or video generation

- **View/frame imputation**: Missing viewpoints or temporal frames are hallucinated by the model, creating the appearance of generalization while the core structure remains memorized

- **Modal adaptation**: A memorized 2D composition is adapted to 3D or temporal constraints, introducing variations while preserving identifying features

*(2) High Prevalence of Templated Content in Training Data.* The second-stage training datasets (Objaverse for MVDream, WebVid-10M for LaVie) inherently encourage template memorization:

- **WebVid-10M**: Contains stock videos sharing identical templates, with only superficial variations (e.g., different flags)

- **Objaverse**: Includes identical 3D mesh geometries paired with different textures/materials, which are treated as separate training samples but share the same underlying structure

This creates an environment where template memorization is not just common but structurally encouraged by the data. Metrics that rely on diversity or cross-seed variation are particularly vulnerable when memorization and generalization coexist within the same generation, such as:

- Templated videos with temporal dilation/contraction, or minor variations in lighting, camera motion, or object motion.

- 3D generations that memorizes mesh geometry with varied textures.

- Multi-view generations that interpolate between multiple memorized images while maintaining cross-view consistency.

These samples are *memorized* in the sense that they reproduce distinctive, identifying features from training data, but they simultaneously exhibit *generalization* behaviours (variation, diversity, interpolation).

## A.2 FURTHER DETAILS ON HIGH-RISK SAMPLE COLLECTION PIPELINE

We establish a reproducible pipeline to identify high-risk samples from the following datasets with many **near-duplicates** in either or both language and visual modalities, and are thus more prone to memorization. Importantly, while the process is selective in order to focus evaluation on samples most likely to exhibit memorization, it does not intentionally bias towards particular styles, creators, or semantic trends, and the same procedure could be applied to any dataset split.

- **LAION:** We use the established benchmark from Webster (2023), which consists of 500 prompts known to elicit memorized images and 500 non-memorizing prompts. This benchmark has been widely used in prior works (Wen et al., 2023; Jeon et al., 2024) for SD memorization evaluation.

- **Objaverse:** Since models trained on Objaverse use different captioning approaches (Cap3D , concatenated object names and tags, or a mixture of both), this selection process is primarily based on the *visual modality* to ensure consistency across captioning variations. (1) *Stratification* using LVIS classes and metadata tags associated with high memorization risk (e.g., popular cultural content); (2) *Geometric similarity detection* within high-risk clusters using objective mesh descriptors (volume, vertex count, surface area) and Wasserstein-1 distance matching, targeting the common pattern where identical meshes are paired with different textures; (3) *Validation* via SSIM-based and feature-based matching with training asset renderings, followed by manual confirmation to remove false positives.

- **WebVid-10M:** Since WebVid-10M exhibits high caption-video agreement (similarity in the language modality is strongly correlated with similarity in the visual modality), we initiate this process from the *language/caption modality* to efficiently identify near-duplicate content. (1) *Caption clustering*: Filter by normalized Levenshtein distance ($>0.85$) to detect near-duplicate textual descriptions, which yields 24,781 initial clusters with 408,670 samples; (2) *Visual diversity filtering*: to remove clusters with high intra-cluster CLIP feature variance, ensuring retained clusters are visually similar and thus prone to being memorized; (3) *Content filtering*: to exclude abstract (kaleidoscope-like) or texture-like clips that do not have strong semantic association with their captions; (4) *Manual confirmation* to verify automated selections.

**Ground Truth Validation for Multi-Modal Memorization**. Establishing reliable ground truth labels for memorization requires a rigorous methodology to minimize labeling errors. Our approach consists of a two-stage pipeline: an automated candidate selection phase followed by manual verification. The automated first pass is designed to systematically identify a comprehensive set of potential memorization events. For 3D assets from Objaverse, we render each training item from eight azimuth angles, while for videos from WebVid10M, we undersample each clip into sixteen keyframes. We then compute similarity scores between these references and the generated content. This process utilizes a dual-metric framework to capture both structural correspondence (SSIM) and semantic alignment (CLIP embeddings). To maximize recall and ensure no potential cases are missed, candidates are flagged for manual review if they surpass a lenient, pre-determined threshold on either of the similarity metrics. Furthermore, by taking the maximum similarity score across all view-to-view or frame-to-frame comparisons, our method remains sensitive to partial reproductions and temporal transformations (e.g. temporal reversal, speed changes, zooming in), which are common signatures of memorization. Finally, all automatically identified candidates are subjected to manual validation involving multiple human annotators, ensuring the reliability and accuracy of the final ground truth labels.

## B EXTENDED BACKGROUND AND RELATED WORKS

### B.1 MEMORIZATION IN DIFFUSION MODELS.

Memorization behaviors in visual diffusion models have been extensively studied primarily through extraction attacks (EA) and membership inference attacks (MIA). EAs (Carlini et al., 2021; Somepalli et al., 2023a; Carlini et al., 2023; Qu et al., 2023; Leotta et al., 2023; Naik & Nushi, 2023; Zhang et al., 2024; Li et al., 2024d; Webster et al., 2023; Webster, 2023; Liao, 2022; Chen et al., 2025b; Wu et al., 2024b; Daras et al., 2024; Wen et al., 2025) show that both specific and implicit user prompts can elicit content closely resembling training data, including copyrighted or sensitive visual content. MIAs further probe memorization risks by determining whether particular inputs were seen during training. These include white-box methods that exploit gradients and internal loss dynamics (Hu & Pang, 2023; Pang et al., 2023; Tang et al., 2023; Kong et al., 2023) and black-box approaches that analyze statistical properties of outputs (Wu et al., 2022; Fu et al., 2024; Dubiński et al., 2024). Beyond these direct attack methods, similarity retrieval techniques employ feature extraction and distance metrics to identify matches between generated content and training data (Radford et al., 2021; Zhou et al., 2023; Rahman et al., 2024), detecting both content and stylistic replication (Casper et al., 2023; Somepalli et al., 2024). Additional strategies include watermarking approaches that embed forensic signals in training data (Wang et al., 2023b; Cui et al., 2023; Luo et al., 2023; Asnani et al., 2024) and personalized replication studies through subject-specific fine-tuning (Ruiz et al., 2023; Ma et al., 2023), which raise important ethical and attribution questions (Shan et al., 2023; Liu et al., 2024).

**Root Cause Analysis.** Insufficient training diversity (Gu et al., 2023) and data duplication (Somepalli et al., 2023b; Li et al., 2024a) have been identified as primary drivers of memorization. Other factors such as prompt specificity (Naseh et al., 2023; Chen et al., 2024a), out-of-distribution characteristics (Janolkar, 2023), deterministic samplers (Yi et al., 2023), and model capacity (Peebles & Xie, 2023; Chen et al., 2024a) also influence memorization extent. New evaluation metrics such as Feature Likelihood Divergence (Jiralerspong et al., 2023; Jagielski et al., 2022) and creativity measures have been introduced to quantify various memorization aspects. Theoretical investigations span information theory (Yi et al., 2023; Li et al., 2023b), cognitice science (Pham et al., 2025), harmonic analysis (Kadkhodaie et al., 2024), geometric (Wang et al., 2024a; Kamkari et al., 2024; Jeon et al., 2024; Ross et al., 2025; Buchanan et al., 2025), and data attribution frameworks (Georgiev et al., 2023; Liu et al., 2025; Taghanaki & Lambourne, 2024; Wang et al., 2024b).

While this body of work provides valuable insights into memorization mechanisms, evaluation efforts have predominantly focused on standard text-to-image architectures, particularly Stable Diffusion variants. Notable exceptions include recent investigations of medical imaging modalities (Rahman et al., 2024; Dar et al., 2023; 2024b; 2025; 2024a) and video diffusion models (Zhao et al., 2024b). However, systematic evaluation of memorization detection methods across the increasingly prevalent multi-stage, multi-modal training protocols that characterize modern diffusion systems

remains limited. This represents an important gap as these complex training regimes may fundamentally alter the memorization patterns that detection methods were designed to identify.

## B.2 MEMORIZATION DETECTION

We briefly review SoTA reference-free memorization metrics used in our evaluation, categorized into three families: score-based, diversity-based, and cross-attention-based.

### B.2.1 SCORE-BASED METRICS

**Noise Difference Norm (NDN).** Wen et al. (2023) proposes to quantify how text conditioning overpowers initial noise influence during generation as a proxy for memorization. For text prompt embedding $e_p$ and null embedding $e_\emptyset$:

$$D = \frac{1}{T} \sum_{t=1}^{T} \left\| p_\theta(x_t, e_p) - p_\theta(x_t, e_\emptyset) \right\|_2 \tag{5}$$

Larger values indicate prompt dominance over noise, signaling memorization. Huang et al. (2025) also demonstrates this term's utility in frequency-selective controllable editing tasks. Chen et al. (2025a) have proposed a localized variant that masks the standard noise difference norm with the attention map of the final token.

**Sharpness via Hessian Eigenvalue Difference (HED).** Jeon et al. (2024) explore the geometry of the model's learned probability distribution, where memorized samples correspond to sharp peaks in the probability landscape. They approximate differences in Hessian eigenvalue magnitudes between conditional and unconditional score functions using directional finite differences. For latents $x_t$ at timestep $t$, computes a normalized perturbation $\Delta x = \delta \cdot \frac{s_\theta(x_t, c) - s_\theta(x_t, \emptyset)}{||s_\theta(x_t, c) - s_\theta(x_t, \emptyset)||}$ with $\delta = 10^{-3}$ in the CFG discrepancy direction, then measures the response magnitudes $||s_\theta(x_t + \Delta x, c) - s_\theta(x_t, c)||$ and $||s_\theta(x_t + \Delta x, \emptyset) - s_\theta(x_t, \emptyset)||$. Under the local Gaussian assumption, this approximates the squared eigenvalue differences $\sum_i \frac{(\lambda_i - \lambda_{i,c})^2}{\lambda_{i,c}}$ where $\lambda_i, \lambda_{i,c}$ are eigenvalues of the unconditional and conditional Hessian matrices $H_u(x_t), H_c(x_t)$. Large magnitude differences indicate regions where conditioning creates sharp, localized probability peaks characteristic of memorized content.

**Bright Ending (BE).** (Chen et al., 2025a) identifies memorized samples through unusually high attention on final prompt tokens, reflecting collapsed focus onto trigger concepts rather than distributed feature composition. This can be considered a spatially aware version of NDN where the noise difference maps are multiplied by a memorization mask extracted via BE:

$$LD = \frac{1}{T} \sum_{t=1}^{T} \left\| (\varepsilon_\theta(x_t, e_p) - \varepsilon_\theta(x_t, e_\phi)) \circ \mathbf{m} \right\|_2 \Big/ \left( \frac{1}{N} \sum_{i=1}^{N} m_i \right) \tag{6}$$

where $N$ is the number of elements in the mask $\mathbf{m}$. Attention scores are directly as weights. The result is normalized by the mean of the attention weights $\mathbf{m}$.

**InvMM.** Ma et al. (2025) propose an inversion-based approach that characterizes memorization through the tractability of mapping a data sample back to the Gaussian prior. The core premise is that memorized samples are easier to invert into a high-likelihood latent region compared to generalized samples. InvMM quantifies this by optimizing a variational distribution $q_\phi(z|x)$ to reconstruct the target image $x$, and then measuring the KLD from the standard prior:

$$M_{\text{Inv}}(x) = \min_\phi D_{\text{KL}}(q_\phi(z|x), ||, \mathcal{N}(0, I)). \tag{7}$$

$p$**-Laplace.** Brokman et al. (2025) introduce a generalization of curvature analysis using the $p$-Laplacian operator to identify memorization. Similarly to HED (which relates to the 2-Laplacian), the $p$-Laplace metric allows for tuning the sensitivity to the gradient magnitude, offering a non-linear measure of how the probability density concentrates around a sample.

$$\Delta_p \log p(x) = \nabla \cdot \left( |\nabla \log p(x)|^{p-2} \nabla \log p(x) \right). \tag{8}$$

### B.2.2 DIVERSITY-BASED MEOMRIZATION METRICS

Both Carlini et al. (2023) and Hintersdorf et al. (2024) propose to measure diversity across different seeds for the same prompt, where low diversity typically indicates memorization tendencies.

**SSIM of Noise Differences.** Hintersdorf et al. (2024) observe that a model's denoising process is "seed-agnostic" or consistent across seeds for memorized prompts. Let $x_T$ be the initial noise image for prompt $y$. Compute the *noise-difference image*

$$\delta = p_\theta(x_T, T, y) - x_T.$$

Generate $\delta^{(i)}$ and $\delta^{(j)}$ from two seeds $i$ and $j$, then evaluate

$$\text{SSIM}\left(\delta^{(i)}, \delta^{(j)}\right) = \frac{(2\mu_i\mu_j + C_1)(2\sigma_{ij} + C_2)}{(\mu_i^2 + \mu_j^2 + C_1)(\sigma_i^2 + \sigma_j^2 + C_2)}. \tag{9}$$

A higher score indicates a seed-insensitive, hence memorised trajectory.

**Median SSCD.** For images $\{I_1, I_2, \ldots, I_n\}$ generated from the same prompt using different seeds, Hintersdorf et al. (2024) computed pairwise cosine similarities between SSCD embeddings. Let $\phi(I_i)$ denote the SSCD embedding of image $I_i$, the similarity between the two images is:

$$\text{sim}(I_i, I_j) = \frac{\phi(I_i) \cdot \phi(I_j)}{\|\phi(I_i)\|\|\phi(I_j)\|} \tag{10}$$

The diversity metric is the median of all pairwise similarities:

$$\text{Diversity}_{\text{SSCD}} = \text{median}\left\{\text{sim}(I_i, I_j) : 1 \leq i < j \leq n\right\} \tag{11}$$

**Tiled $\ell_2$ (TL2).** Carlini et al. (2023) divide each image into non-overlapping $128 \times 128$ tiles and compute pairwise distances. For images $I_i$ and $I_j$, let $T_k^{(i)}$ and $T_k^{(j)}$ denote their $k$-th tiles respectively. The tiled distance between two images is:

$$d_{\text{tiled}}(I_i, I_j) = \frac{1}{K} \sum_{k=1}^{K} \|T_k^{(i)} - T_k^{(j)}\|_2 \tag{12}$$

where $K$ is the total number of tiles. The diversity metric is the minimum pairwise distance:

$$\text{Diversity}_{\ell_2} = \min\left\{d_{\text{tiled}}(I_i, I_j) : 1 \leq i < j \leq n\right\} \tag{13}$$

**Potential Failure Modes in Novel DMs.** Diversity-based metrics are relatively more robust across varied training protocols; however, a noticeable shrinkage in separability still occurs. We identified two potential causes:

(1) Data-induced ambiguity: As noted in Sec. 3.1, the datasets themselves often contain clusters of near-duplicates. For instance, WebVid10M has stock videos sharing a scene template with minor differences, and Objaverse pairs identical meshes with different textures. A model correctly learns to reproduce this entire cluster of variations. Consequently, a memorized generation may exhibit high diversity that simply reflects the concept's inherent variance in the training data.

(2) Training-induced variance: Multi-modal domains have many more degrees of freedom and often lack a single canonical representation. Training protocols exploit this via data augmentation; for instance, 3D models are trained with varied camera angles (and often without canonical azimuths), while video models use temporal augmentations like shifting or flipping. This teaches the model to reproduce a memorized concept with these variations across seeds, artificially inflating the diversity of memorized samples and making them appear less stable and more like novel generations.

### B.2.3 CROSS-ATTENTION-BASED METRICS

**Cross-Attention Entropy (CAE).** (Ren et al., 2024) This approach quantifies the dispersion of cross-attention scores to detect memorization. Two metric variants are introduced.

The first global variant, CAE-$D$, captures the phenomenon where memorized samples maintain dispersed attention on summary and prompt tokens during the later denoising steps ($t \rightarrow 0$), whereas non-memorized samples concentrate attention on the beginning token. This is calculated as:

$$D = \frac{1}{T_D} \sum_{t=0}^{T_D - 1} E_t + \frac{1}{T_D} \sum_{t=0}^{T_D - 1} |E_t^{summary} - E_T^{summary}|$$

where $T_D$ represents the number of final steps considered (e.g., $T/5$), $E_t$ is the standard Shannon entropy, and $E^{summary}$ is the entropy calculated specifically on summary tokens.

The second local variant, CAE-$E_{t=T}^l$, computes the entropy the very first diffusion step ($t = T$) as

$$E_{t=T}^l = \sum_{i=1}^{N} -\overline{a}_i^l \log(\overline{a}_i^l)$$

where $\overline{a}_i^l$ denotes the averaged attention score of the $i$-th token on the $l$-th layer at step $T$. Although the polarity of this metric may vary across diffusion timesteps $t$, model layers $l$, and attention heads (such that higher values in some layers correspond to memorized content and lower values to unmemorized content) this metric generally preserves strong separability between memorized and unmemorized samples across most layers, serving as a robust indicator of memorization.

Certain layers exhibit greater discriminative power in distinguishing memorized from non-memorized content, with the optimal layer being dataset-dependent. Following the original work, we adopt the local variant CAE-$E$ at $t = T$ using the 4$^{\text{th}}$ layer for evaluation, while also reporting results for the best-performing layer.

**Bright Ending (BE).** (Chen et al., 2025a) also belongs to this category.

**Potential Failure Modes in Novel DMs.** The performance degradation of attention-based metrics can likely be attributed to architectural and representational shifts. These metrics assume memorization is linked to certain attention patterns (e.g. concentrated, low-entropy pattern) on specific prompt tokens (e.g. the last token). However, many multi-stage models are built by fundamentally altering their attention mechanisms to support new modalities. This includes (1) architectural changes, such as MVDream inflating 2D self-attention layers into 3D cross-view ones or LaVie incorporating temporal self-attention; (2) the injection of additional conditioning signals like camera embeddings, task-level prefixes Kant et al. (2024), task-level postfixes Shi et al. (2023), or other geometric inputs. This enriched conditioning complicates the attention landscape.

### B.3 Memorization Mitigation.

In response to growing concerns about memorization behaviors, interventions to mitigate memorization risks have been developed across various stages of the model development lifecycle. These approaches can be broadly categorized into preemptive and post-training strategies.

**Preemptive Mitigation.** Data-centric approaches aim to limit exposure to replicable content before training begins, including semantic deduplication (Abbas et al., 2023) and copyright-safe dataset curation (Gokaslan et al., 2023). Training-time architectural strategies include compositionally isolated training (Golatkar et al., 2023), despecification guidance (Chen et al., 2024a), and replication-aware architectures like LoyalDiffusion (Li et al., 2024b).

**Post-Training Mitigation.** Post-training approaches offer several practical advantages: they can be applied to already-deployed models, do not require access to original training data, and can target specific problematic content without full retraining. Early unlearning methods like Concept Ablation (Kumari et al., 2023), Forget-Me-Not (Zhang et al., 2023), and ErasedDiff (Wu et al., 2024a) modify internal model representations to erase learned associations. Scalable approaches enable batch concept removal (Fan et al., 2023; Zhao et al., 2024a; Hong et al., 2024; Lu et al., 2024), while recent work explores regularization-based suppression (Ni et al., 2023) and multi-concept editing (Xiong et al., 2024; Gandikota et al., 2024). Inference-time techniques provide additional flexibility through perturbation, attention reweighting (Ren et al., 2024), token masking (Chen et al., 2025a), neuron suppression (Hintersdorf et al., 2024; Chavhan et al., 2024), and guided sampling

methods (Li et al., 2024c; Dong et al., 2023). While these approaches require no model modification, their effectiveness can vary with prompt complexity and adversarial inputs.

Given their practical advantages, post-training mitigation methods represent a critical component of modern diffusion model safety pipelines. This motivates our focus on evaluating the robustness of reference-free detection metrics as evaluation tools for post-training memorization mitigation.

Table 6: Post-Training Memorisation Mitigation Methods evaluated in this work.

| Acronym | Method | Category |
|---|---|---|
| RTA | Random Token Addition (Somepalli et al., 2023b) | Perturbation, token space |
| RNA | Random Number Addition (Somepalli et al., 2023b) | Perturbation, token space |
| CWR | Caption Word Repetition (Somepalli et al., 2023b) | Perturbation, token space |
| GNI | Gaussian Noise Injection (Somepalli et al., 2023b) | Perturbation, embedding space |
| Wen | Wen's Mitigation (Wen et al., 2023) | Perturbation, embedding space |
| SAIL | Sharpness-Aware InitiaLization (Jeon et al., 2024) | Initial Noise Optimization |
| BE | Bright Ending Mitigation (Chen et al., 2025a) | Attention Adjustment |
| CAR | Cross Attention Reweighting (Ren et al., 2024) | Attention Adjustment |
| NeMo | Neurons responsible for meMorization (Hintersdorf et al., 2024) | Model Editing |
| SubPrune | Subspace Pruning (Chavhan et al., 2024) | Model Editing |
| UCE | Unified Concept Editing (Gandikota et al., 2024) | Model Editing |
| AMG | Anti-Memorization Guidance (Chen et al., 2024a) | Guided Sampling |

### B.3.1 MEMORIZATION MITIGATION METHODS AND IMPLEMENTATION DETAILS.

We briefly reviewed the memorization mitigation methods evaluated in Sec. 3.3. Below, we provide a more detailed description of their mechanisms and our implementation.

The first four of the evaluated methods are simple, metric-agnostic perturbations introduced by (Somepalli et al., 2023b). These methods do not require knowledge of the detection metric and instead aim to disrupt memorization by altering the input prompt or embedding.

**Random Token Addition (RTA).**    This metric-agnostic perturbation method disrupts memorized text-image associations by injecting random tokens into input prompts. We randomly select token IDs between 1000-40000 from the tokenizer vocabulary and insert them at arbitrary positions within the prompt string, adding 4 random tokens per prompt by default.

**Random Number Addition (RNA).**    RNA creates distribution shift from training data by appending random numerical values to prompts. The implementation generates random integers between 0 and 1,000,000 and inserts them at random positions, adding 10 random numbers per prompt to effectively move prompts out of the training distribution.

**Caption Word Repetition (CWR).**    This approach alters token frequency distribution by repeating existing words within prompts. We split prompts into words, randomly select from the existing vocabulary, and insert them at random positions, performing 10 repetitions per prompt to significantly change prompt structure while maintaining semantic content.

**Gaussian Noise Injection (GNI).**    GNI disrupts memorized embedding patterns by adding Gaussian noise directly to text embeddings. We apply noise $\epsilon \sim \mathcal{N}(0, \sigma^2)$ with standard deviation $\sigma = 0.5$ to the text embedding tensor after encoding but before cross-attention computation.

**Wen's Perturbation.**    This metric-aware approach perturbs the text embedding in a targeted manner (Wen et al., 2023). It is an optimization-based technique that uses gradient descent to iteratively modify the embedding. The objective is to minimize the global noise difference norm ($D$), which is the core signal used by the NDN detection metric. Our implementation uses Adam optimizer and performs a fixed number of optimization steps on the embedding minimizing the loss $L = \|\epsilon_\theta(x_t, t, c) - \epsilon_\theta(x_t, t, \emptyset)\|_2$ before the main denoising process begins. $c$ represents the perturbed embedding. This metric-aware approach optimizes text embeddings by minimizing the magnitude of conditional noise predictions.

**Sharpness-Aware Initialization (SAIL).**    SAIL Jeon et al. (2024) is an initial noise optimization technique that seeks to find a starting latent vector $x_T$ that is situated in a "non-sharp" region of the probability landscape, thus avoiding pathways that lead to memorized outputs. SAIL optimizes the initial noise $x_T$ to find latents that lead to less memorized outputs. Implementation follows Algorithm 2 from the original paper, using finite differences to estimate score function sharpness. The optimization objective is $L = \|s_\delta(x_T + \delta \cdot \frac{s_\delta(x_T)}{\|s_\delta(x_T)\|}) - s_\delta(x_T)\|^2 + \alpha\|x_T\|^2$ where $s_\delta = \epsilon_\theta(x_t, t, c) - \epsilon_\theta(x_t, t, \emptyset)$.

**Bright Ending (BE) Mitigation.**    BE Chen et al. (2025a) uses attention maps to identify regions likely to contain memorized content and minimizes noise differences in those areas. Implementation collects cross-attention maps from down-sampling blocks during denoising, averages across attention heads, and creates masks highlighting high-attention regions. The optimization objective is $L = \frac{\|\text{mask} \odot (s_\delta)\|_2}{\text{mean(mask)}+10^{-6}}$ where $\odot$ denotes element-wise multiplication.

**Cross-Attention Reweighting (CAR).**    This attention-adjustment method operates during inference by directly manipulating cross-attention scores (Ren et al., 2024). It first identifies tokens that are likely triggers for memorization by analyzing their attention concentration (entropy). During generation, it then suppresses the attention scores corresponding to these trigger tokens, effectively reducing their influence on the final output without altering the model's weights.

**NeMo.**    This model-editing technique deactivates memorization-responsible neurons at inference time Hintersdorf et al. (2024). For memorized prompts, we identify candidate neurons in cross-attention value-projection layers based on outlier activations (z-score against non-memorized baseline) and high activation levels (top-k). We iteratively expand the suppression set until memorization score (max pairwise SSIM of initial noise differences) falls below a threshold.

**Subspace Pruning (SubPrune).**    This is an offline model-editing approach that permanently removes weights deemed responsible for memorization (Chavhan et al., 2024). It operates on the principle that memorized samples share a common activation subspace. This method identifies weight subspaces in FFN layers critical for memorization and sets them to zero. Our implementation targets the second linear layer in feed-forward networks `ff.net.2`, computes saliency scores $S = |W| \cdot \|H\|$ where $W$ are weights and $H$ are activations, and prunes weights where memorized activations exceed null prompt activations. Sparsity level set to 0.1% of total weights in our evaluation.

**Unified Concept Editing (UCE).**    UCE is an offline model-editing technique that modifies the weights of key and value projection matrices in cross-attention layers to erase concepts (Gandikota et al., 2024). UCE uses a closed-form update equation to solve for new weights. This update aims to map an "erase" concept embedding to a "guide" concept's output, while optionally preserving the model's behavior on other specified concepts. UCE modifies cross-attention weights using closed-form updates to erase specific concepts while preserving others. Implementation targets `to_k` and `to_v` layers in cross-attention blocks, applying the update rule $W_{\text{new}} = (\lambda W_{\text{old}} + \sum_i \alpha_i v_i^* c_i^T)(\lambda I + \sum_i \alpha_i c_i c_i^T)^{-1}$ where $c_i$ are concept embeddings and $v_i^*$ are target outputs. Default regularization $\lambda = 0.5$ with erase/preserve scales of 1.0.

**Anti-Memorization Guidance (AMG):**    Chen et al. (2024a) have proposed a guided sampling method that prevents memorization by steering the generation process away from training data exemplars. AMG uses nearest neighbour search to detect potential memorization during sampling and steers generation away from training data. Implementation employs CLIP embeddings to build a training data index, computes cosine similarity between generated images and training examples, and applies classifier guidance when similarity exceeds threshold. The guidance term modifies the score function to reduce likelihood of generating near-duplicates of training images.

### B.3.2    MEASURING CFG DISCREPANCY-BASED METRICS WITH AMG.

To correctly measure various flacours of CFG discrepancy-based metrics on guided trajectories such as AMG (Chen et al., 2024a), we define an *effective conditional prediction*, $\tilde{p}_\theta(x_t, e_p)$, that accounts for our additional guidance term, $g_{\text{AMG}}$.

The final noise prediction, $\tilde{\epsilon}_\theta$, which incorporates both the standard Classifier-Free Guidance (CFG) and our AMG term, is given by:

$$\tilde{\epsilon}_\theta(x_t, e_p, e_\emptyset) = p_\theta(x_t, e_\emptyset) + w \cdot (p_\theta(x_t, e_p) - p_\theta(x_t, e_\emptyset)) + g_{\text{AMG}} \tag{14}$$

where $w$ is the guidance scale.

Our goal is to find an effective conditional prediction $\tilde{p}_\theta(x_t, e_p)$ that, when substituted into the standard CFG formula, yields this same $\tilde{\epsilon}_\theta$:

$$\tilde{\epsilon}_\theta(x_t, e_p, e_\emptyset) = p_\theta(x_t, e_\emptyset) + w \cdot (\tilde{p}_\theta(x_t, e_p) - p_\theta(x_t, e_\emptyset)) \tag{15}$$

By equating (14) and (15), we can solve for $\tilde{p}_\theta(x_t, e_p)$. The unconditional term $p_\theta(x_t, e_\emptyset)$ cancels, leaving:

$$w \cdot (\tilde{p}_\theta(x_t, e_p) - p_\theta(x_t, e_\emptyset)) = w \cdot (p_\theta(x_t, e_p) - p_\theta(x_t, e_\emptyset)) + g_{\text{AMG}}$$

$$\tilde{p}_\theta(x_t, e_p) - p_\theta(x_t, e_\emptyset) = (p_\theta(x_t, e_p) - p_\theta(x_t, e_\emptyset)) + \frac{g_{\text{AMG}}}{w}$$

This yields the expression for the effective conditional prediction, which correctly isolates the guidance term as an additive modification to the original conditional prediction, scaled inversely by the guidance weight:

$$\tilde{p}_\theta(x_t, e_p) = p_\theta(x_t, e_p) + \frac{g_{\text{AMG}}}{w} \tag{16}$$

## C   FURTHER DETAILS ON ASSUMPTIONS AND PROXY MEASUREMENTS

This section supplements Sec. 4.3 by providing more details about the connection between assumptions and geometric framework and implementation details of the diagnostic measurements.

### C.1   HOW ASSUMPTIONS SUPPORT THE GEOMETRIC FRAMEWORK

Various score-based metrics can be connected through a chain of approximate equivalences, which are also linked to their efficacy and generalizability.

$$\underbrace{\mathbb{E}[\|s_\theta(x, c) - s_\theta(x)\|^2]}_{\text{NDN, BE, DiffSSIM}} \overset{\text{A1}}{\approx} \underbrace{\mathbb{E}[\|s(x, c) - s(x)\|^2]}_{\text{True Score Difference}} \overset{\text{A2}}{\approx} \underbrace{\text{tr}[(H - H_c)^2 \Sigma_c]}_{\text{Curvature Trace}} \overset{\text{A2+A4}}{\approx} \underbrace{\sum_i \frac{(\lambda_i - \lambda_{i,c})^2}{\lambda_{i,c}}}_{\text{Eigenvalue Gaps (HED)}} \tag{17}$$

Consider a diffusion process along timesteps $t \in [0, T]$ from clean data $x_0$ to noisy latent $x_T$, where

- $p_t(x_t)$: unconditional distribution of noisy latent $x_t$
- $p_t(x_t|c)$: conditional distribution given prompt/conditioning $c$
- $s_t(x_t) = \nabla_{x_t} \log p_t(x_t)$: unconditional score function (true)
- $s_t(x_t, c) = \nabla_{x_t} \log p_t(x_t|c)$: conditional score function (true)
- $s_\theta(x_t)$, $s_\theta(x_t, c)$: model's estimated score functions

**Geometric Quantities.**   The Hessian of the log-density encodes local curvature:

$$H_t(x_t) = \nabla_{x_t}^2 \log p_t(x_t) = \left[\frac{\partial^2 \log p_t}{\partial x_i \partial x_j}\right]_{i,j=1}^d, \tag{18}$$

$$H_{t,c}(x_t) = \nabla_{x_t}^2 \log p_t(x_t|c). \tag{19}$$

Under Gaussian assumptions, these relate to covariance matrices. For $x \sim \mathcal{N}(\mu, \Sigma)$:

$$\log p(x) = -\frac{1}{2}(x - \mu)^\top \Sigma^{-1}(x - \mu) + \text{const}, \tag{20}$$

giving:

$$\nabla_x \log p(x) = -\Sigma^{-1}(x - \mu), \tag{21}$$

$$\nabla_x^2 \log p(x) = -\Sigma^{-1}. \tag{22}$$

A core detection signal is the difference between conditional and unconditional scores:

$$\Delta s_t(x_t, c) = s_t(x_t, c) - s_t(x_t). \tag{23}$$

The NDN metric (Wen et al., 2023) aggregates this over timesteps:

$$\text{NDN} = \frac{1}{T} \sum_{t=1}^{T} \|\epsilon_\theta(x_t, t, c) - \epsilon_\theta(x_t, t, \emptyset)\|^2, \tag{24}$$

where $\epsilon_\theta$ denotes the noise prediction network. Since $\epsilon_\theta$ and $s_\theta$ are linearly related via:

$$s_\theta(x_t, c) = -\frac{\epsilon_\theta(x_t, t, c)}{\sigma_t}, \tag{25}$$

the NDN can be expressed in terms of score differences.

For a Gaussian $x \sim \mathcal{N}(\mu, \Sigma)$:

$$\log p(x) = -\frac{1}{2}(x - \mu)^\top \Sigma^{-1}(x - \mu) + \text{const}, \tag{26}$$

which gives:

$$s(x) = \nabla_x \log p(x) = -\Sigma^{-1}(x - \mu), \tag{27}$$

$$H(x) = \nabla_x^2 \log p(x) = -\Sigma^{-1}. \tag{28}$$

**Quadratic Forms.** For $x \sim \mathcal{N}(\mu, \Sigma)$ and symmetric matrix $A$:

$$\mathbb{E}[(x - \mu)^\top A(x - \mu)] = \text{tr}(A\Sigma). \tag{29}$$

*Proof:* Let $z = x - \mu \sim \mathcal{N}(0, \Sigma)$. Then $\mathbb{E}[z^\top A z] = \sum_{i,j} A_{ij}\mathbb{E}[z_i z_j] = \sum_{i,j} A_{ij}\Sigma_{ij} = \text{tr}(A\Sigma)$.

**Assumptions Revisited.** We re-state all six key assumptions that will be invoked in subsequent proofs.

**Assumption (A1)** (Unbiased Score Estimation). *For all $x_t$, $c$, and $t$:*

$$\mathbb{E}_\theta[s_\theta(x_t, c) - s_t(x_t, c)] = 0, \tag{30}$$

*where the expectation is over the stochasticity in the trained model (if any). Equivalently, $s_\theta$ is an unbiased estimator of the true score.*

**Assumption (A2)** (Gaussian Local Structure). *At relevant timesteps $t \in \mathcal{T}$, the distributions are approximately Gaussian:*

$$p_t(x_t) \approx \mathcal{N}(\mu_t, \Sigma_t), \tag{31}$$

$$p_t(x_t|c) \approx \mathcal{N}(\mu_{t,c}, \Sigma_{t,c}), \tag{32}$$

*where the approximation error is small enough that Eqs. 27–28 hold to sufficient precision.*

**Assumption (A3)** (Sharpness Persistence Across Timesteps). *Let $\mathcal{E}_t = \{\lambda_i^{(t)}\}_{i=1}^d$ denote the spectrum (ordered eigenvalues) of $H_t(x_t)$ or $H_{t,c}(x_t)$. The distinguishability between memorized and non-memorized samples, as measured by differences in $\mathcal{E}_t$, persists over the interval $t \in [t_{\min}, T]$ used for detection.*

**Assumption (A4)** (Covariance Commutativity and Mean Equality). *For relevant timesteps: The covariance matrices commute: $\Sigma_t \Sigma_{t,c} = \Sigma_{t,c} \Sigma_t$. The means coincide: $\mu_t = \mu_{t,c}$.*

**Assumption (A5)** (Mean-Field Gaussian Prior). *The initial latent follows a standard normal distribution:*

$$x_T \sim \mathcal{N}(0, I_d), \tag{33}$$

*where $I_d$ is the $d$-dimensional identity matrix.*

**Assumption (A6)** (Boundary Regularity). *The score function satisfies regularity conditions: (1) Vanishing boundary condition: $\lim_{\|x\| \to \infty} p(x)s(x) = 0$; (2) Finite second moment: $\mathbb{E}[\|s(x)\|^2] < \infty$.*

### C.1.1 ASSUMPTION A1: BRIDGING MODEL OUTPUTS TO THEORY

Assumption (A1) enables the first approximation in Eq. 17. All theoretical results (Lemmas 1–3) are derived for *true* score functions $s(x)$. To apply these results to practical metrics computed from model outputs $s_\theta(x)$, we require:

Under Assumption (A1):

$$\mathbb{E}_\theta[\|s_\theta(x,c) - s_\theta(x)\|^2] \approx \mathbb{E}[\|s(x,c) - s(x)\|^2]. \tag{34}$$

Biased score estimations (e.g., due to non-standard training objectives) weaken the connection between measured quantities and the theoretical geometric quantities.

### C.1.2 ASSUMPTION A2: ENABLING ANALYTICAL TRACTABILITY

((A2)) is a primary assumptions that enables approximate equivalence in Eq. 4. In particular:

1. Explicit forms for score and Hessian (Eq. 27–28)

2. Closed-form expectations via Eq. 29

3. The second and third approximations in Eq. 17

These three key lemmas rely on (A2) in the following ways:

**Lemma 1: Relating Score Norm to Hessian Trace**

**Lemma 1** (Score Norm and Hessian Trace). *For $x \sim \mathcal{N}(\mu, \Sigma)$:*

$$\mathbb{E}[\|s(x)\|^2] = -\operatorname{tr}(H(x)) = \operatorname{tr}(\Sigma^{-1}). \tag{35}$$

*Proof.* **Step 1 [Invokes A2]:** Under Gaussianity, $s(x) = -\Sigma^{-1}(x - \mu)$ and $H(x) = -\Sigma^{-1}$.

**Step 2:** Compute squared norm:

$$\|s(x)\|^2 = (x - \mu)^\top \Sigma^{-2} (x - \mu). \tag{36}$$

**Step 3:** Apply Eq. 29 with $A = \Sigma^{-2}$:

$$\mathbb{E}[\|s(x)\|^2] = \operatorname{tr}(\Sigma^{-2}\Sigma) = \operatorname{tr}(\Sigma^{-1}) = -\operatorname{tr}(H(x)). \tag{37}$$

$\square$

**Interpretation.** This lemma converts measurable first-order information ($\|s(x)\|^2$) into second-order geometric information ($\operatorname{tr}(H)$), revealing the distribution's average curvature.

**Lemma 2: Higher-Order Curvature**

**Lemma 2** (Score-Hessian Product Norm). *For $x \sim \mathcal{N}(\mu, \Sigma)$:*

$$\mathbb{E}[\|H(x)s(x)\|^2] = -\operatorname{tr}((H(x))^3) = \operatorname{tr}(\Sigma^{-3}). \tag{38}$$

*Proof.* **Step 1 [Invokes A2]:** $H(x) = -\Sigma^{-1}$, $s(x) = -\Sigma^{-1}(x - \mu)$.

**Step 2:** $H(x)s(x) = \Sigma^{-2}(x - \mu)$, so $\|H(x)s(x)\|^2 = (x - \mu)^\top \Sigma^{-4}(x - \mu)$.

**Step 3:** Apply Eq. 29:

$$\mathbb{E}[\|H(x)s(x)\|^2] = \operatorname{tr}(\Sigma^{-3}) = -\operatorname{tr}(H^3). \tag{39}$$

$\square$

**Interpretation.** This measures how rapidly the score changes, quantifying "sharpness" of the probability landscape.

**Lemma 3: CFG Discrepancy Decomposition**

**Lemma 3** (CFG Discrepancy Decomposition). *For $x \sim \mathcal{N}(\mu, \Sigma)$ and $x|c \sim \mathcal{N}(\mu_c, \Sigma_c)$:*

$$\mathbb{E}_{x \sim p(x|c)}[\|s(x,c) - s(x)\|^2] = \|H(\mu - \mu_c)\|^2 + \text{tr}[(H - H_c)^2 \Sigma_c], \tag{40}$$

*where $H = -\Sigma^{-1}$ and $H_c = -\Sigma_c^{-1}$.*

*Proof.* **Step 1 [Invokes A2]:** Under Gaussianity:

$$s(x) = -\Sigma^{-1}(x - \mu), \quad s(x,c) = -\Sigma_c^{-1}(x - \mu_c). \tag{41}$$

**Step 2:** Write the score difference in terms of $(x - \mu_c)$:

$$s(x,c) - s(x) = -\Sigma_c^{-1}(x - \mu_c) + \Sigma^{-1}(x - \mu) \tag{42}$$
$$= -\Sigma_c^{-1}(x - \mu_c) + \Sigma^{-1}(x - \mu_c + \mu_c - \mu) \tag{43}$$
$$= (\Sigma^{-1} - \Sigma_c^{-1})(x - \mu_c) + \Sigma^{-1}(\mu_c - \mu). \tag{44}$$

Define $\Delta H = \Sigma^{-1} - \Sigma_c^{-1} = -(H - H_c)$ and $\Delta\mu = \mu_c - \mu$.

**Step 3:** Compute squared norm:

$$\|s(x,c) - s(x)\|^2 = \|\Delta H(x - \mu_c) + \Sigma^{-1}\Delta\mu\|^2 \tag{45}$$
$$= (x - \mu_c)^\top (\Delta H)^2 (x - \mu_c) + 2(x - \mu_c)^\top \Delta H \Sigma^{-1} \Delta\mu + \|\Sigma^{-1}\Delta\mu\|^2. \tag{46}$$

**Step 4:** Take expectation with $x \sim \mathcal{N}(\mu_c, \Sigma_c)$:

$$\mathbb{E}[\|s(x,c) - s(x)\|^2] \tag{47}$$
$$= \text{tr}[(\Delta H)^2 \Sigma_c] + 0 + \|\Sigma^{-1}(\mu_c - \mu)\|^2 \tag{48}$$
$$= \text{tr}[(H - H_c)^2 \Sigma_c] + \|H(\mu - \mu_c)\|^2, \tag{49}$$

where we used $(\Delta H)^2 = (H - H_c)^2$ and $\Sigma^{-1} = -H$. $\square$

Together with (A2), (A4) establishes the rightmost approximate equivalence in Eq. 17.

### C.1.3 ASSUMPTION A3: ENABLING EARLY DETECTION

Lemmas 1–3 and Proposition C.1.4 provide **static, per-timestep** results. Assumption (A3) extends these to **trajectory-level predictions**.

Let $\mathcal{S}_t$ denote the distribution of geometric signatures (eigenvalue spectra, NDN values, HED values) at timestep $t$. Assumption (A3) requires:

$$\text{Corr}(\mathcal{S}_t^{\text{mem}}, \mathcal{S}_0^{\text{mem}}) \geq \rho_{\min} > 0 \quad \text{for all } t \in [t_{\min}, T]. \tag{50}$$

(A3) provides temporal correlation of geometric signatures, ensuring $t = 50$ measurements predicts whether the final output at $t = 0$ is memorized.

### C.1.4 ASSUMPTION A4: ENABLING EIGENVALUE INTERPRETATION

Assumption (A4) simplifies the trace term in Lemma 3 into interpretable eigenvalue differences, completing the third $\approx$ in Eq. 17.

Under Assumptions (A2) and (A4):

$$\mathbb{E}_{x \sim p(x|c)}[\|s(x,c) - s(x)\|^2] = \sum_{i=1}^{d} \frac{(\lambda_i - \lambda_{i,c})^2}{\lambda_{i,c}}, \tag{51}$$

where $\lambda_i$ and $\lambda_{i,c}$ are eigenvalues of $\Sigma^{-1}$ and $\Sigma_c^{-1}$.

*Proof.* **Step 1 [Invokes A4 - Mean Equality]:** Since $\mu = \mu_c$, the mean term vanishes:

$$\mathbb{E}[\|s(x,c) - s(x)\|^2] = \text{tr}[(H - H_c)^2 \Sigma_c]. \tag{52}$$

**Step 2 [Invokes A4 - Commutativity]:** Since $\Sigma\Sigma_c = \Sigma_c\Sigma$, they share eigenbasis $Q$:

$$\Sigma = Q\text{diag}(1/\lambda_1, \ldots, 1/\lambda_d)Q^\top, \quad \Sigma_c = Q\text{diag}(1/\lambda_{1,c}, \ldots, 1/\lambda_{d,c})Q^\top. \tag{53}$$

**Step 3:** The Hessians are:

$$H = -Q\text{diag}(\lambda_1, \ldots, \lambda_d)Q^\top, \quad H_c = -Q\text{diag}(\lambda_{1,c}, \ldots, \lambda_{d,c})Q^\top. \tag{54}$$

**Step 4:** Compute:

$$(H - H_c)^2 = Q\text{diag}((\lambda_1 - \lambda_{1,c})^2, \ldots, (\lambda_d - \lambda_{d,c})^2)Q^\top. \tag{55}$$

**Step 5:** The trace becomes:

$$\text{tr}[(H - H_c)^2 \Sigma_c] \tag{56}$$

$$= \text{tr}\left[Q\text{diag}((\lambda_i - \lambda_{i,c})^2) \cdot \text{diag}(1/\lambda_{i,c})Q^\top\right] \tag{57}$$

$$= \sum_{i=1}^{d} \frac{(\lambda_i - \lambda_{i,c})^2}{\lambda_{i,c}}. \tag{58}$$

$\square$

(A4) guarantees that:

- Each term $\frac{(\lambda_i - \lambda_{i,c})^2}{\lambda_{i,c}}$ measures the **normalized curvature gap** in direction $i$
- **Memorization signature**: $\lambda_{i,c} \gg \lambda_i$ (conditional distribution is sharper)
- The sum aggregates across all directions, detecting samples with sharp conditional peaks

Without commutativity, the trace does not simplify to eigenvalue differences. Instead:

$$\text{tr}[(H - H_c)^2 \Sigma_c] = \sum_{k,j} (\lambda_k - 1)^2 w_{k,j}, \tag{59}$$

where $w_{k,j}$ are misalignment weights. This requires **generalized eigenvalue analysis**, losing the clean directional interpretation.

### C.1.5   ASSUMPTION A5: ENSURING RELIABLE INITIALIZATION

Assumption (A5) ensures Assumption (A2) holds **most accurately at** $t = T$.

If $x_T \sim \mathcal{N}(0, \Sigma)$ with $\Sigma \neq I$ (e.g., in models with structured priors), the regularizer $\|x_T\|^2$ is misspecified. The correct form would be:

$$\alpha\|x_T\|^2_{\Sigma^{-1}} = \alpha x_T^\top \Sigma^{-1} x_T. \tag{60}$$

### C.1.6   ASSUMPTION A6: PROVIDING TECHNICAL RIGOR

Assumption (A6) serves two purposes:

1. **Enables generalization**: Lemma 1 can be extended beyond Gaussianity
2. **Tolerance bound**: Quantifies permissible deviation from A2

**Lemma 4** (Generalized Score-Hessian Identity). *For $x$ with density $p(x)$ satisfying Assumption (A6):*

$$\mathbb{E}[\|s(x)\|^2] = -\mathbb{E}[\text{tr}(H(x))]. \tag{61}$$

*Proof.* **Step 1:** Write $\mathbb{E}[\|s(x)\|^2] = \sum_i \int s_i(x)^2 p(x)dx$ where $s_i = \partial_i \log p$.

**Step 2:** Use $s_i(x)p(x) = \partial_i p(x)$:

$$\int s_i(x)^2 p(x)dx = \int s_i(x)\partial_i p(x)dx. \tag{62}$$

**Step 3 [Invokes A6]:** Integrate by parts:

$$= \underbrace{[s_i(x)p(x)]_{-\infty}^{\infty}}_{=0 \text{ by A6}} - \int p(x)\partial_i s_i(x)dx = -\mathbb{E}[\partial_i s_i(x)]. \tag{63}$$

**Step 4:** Recognize $\partial_i s_i = H_{ii}$:

$$\mathbb{E}[\|s(x)\|^2] = -\sum_i \mathbb{E}[H_{ii}(x)] = -\mathbb{E}[\mathrm{tr}(H(x))]. \tag{64}$$

$\square$

For non-Gaussian $p(x)$, the approximation error can be estimated by:

$$\epsilon_{\mathrm{Gauss}} = \left| \mathbb{E}[\mathrm{tr}(H(x))] - \mathrm{tr}(-\Sigma^{-1}) \right|. \tag{65}$$

Assumption (A6) ensures this remains bounded via $\mathbb{E}[\|s(x)\|^2] < \infty$.

### C.2 Implementation and Sensitivity Analysis of Diagnostic Measurements

To empirically test the validity of the geometric assumptions listed in Table 1, we implement measurable proxies that translate each assumption into a computable statistic. These diagnostics follow the framework of Jeon et al. (2024) and are designed to check whether the structural conditions required for the equivalence relationships in Eq. (4) remain approximately satisfied in practice. Below, we detail the implementation of each proxy.

Additionally, since these metrics are noisy and have many hyperparameter choices, we also provide sensitivity analysis with respect to measurement-specific hyperparameter choies on SD 1.4.

**(A1) Unbiased Score Estimation.** The assumption requires that the model's estimated score function matches the gradient of the log-density in expectation:

$$\mathbb{E}[s_\theta(x, c) - \nabla_x \log p(x|c)] = 0.$$

We measure *score matching consistency* using Hutchinson's trace estimator:

$$\mathcal{D}_{\mathrm{A1}} = \left( 1 + \left| \nabla \cdot s_\theta(x) + \tfrac{1}{2}\|s_\theta(x)\|^2 \right| \Big/ \|s_\theta(x)\| \right)^{-1}.$$

High values indicate low bias, ensuring the validity of measurements.

Table 7: Proxy Measure Sensitivity Analysis for $\mathcal{D}_{\mathrm{A1}}$. The Hutchinson trace estimator converges with very few random vectors, with negligible gain beyond $n = 5$ and are consistent across timestep selection variants. The choice between Gaussian and Rademacher random vectors has minimal impact.

| Test | Hyperparameter | $\mathcal{D}_{\mathrm{A1}}$ | Test | Hyperparameter | $\mathcal{D}_{\mathrm{A1}}$ |
|------|------|------|------|------|------|
| | $n$=1 | $0.4994 \pm 0.0086$ | RV Distribution | Gaussian | $0.4996 \pm 0.0036$ |
| | $n$=2 | $0.4983 \pm 0.0031$ | | Rademacher | $0.4996 \pm 0.0022$ |
| | $n$=3 | $0.4987 \pm 0.0041$ | | $\{0, 2, \ldots, 48\}$ | $0.4986 \pm 0.0039$ |
| Number of Random Vectors | $n$=5 | $0.4999 \pm 0.0037$ | | $\{47, 48, 49\}$ | $0.4760 \pm 0.0220$ |
| | $n$=10 | $0.5002 \pm 0.0027$ | Timestep Selection | $\{0, 1, 2\}$ | $0.5031 \pm 0.0021$ |
| | $n$=20 | $0.5002 \pm 0.0031$ | | $\{24, 25, 26\}$ | $0.4993 \pm 0.0040$ |
| | $n$=30 | $0.4991 \pm 0.0030$ | | $\{12, 25, 37\}$ | $0.4988 \pm 0.0037$ |
| | $n$=50 | $0.4993 \pm 0.0024$ | | $\{8, 16, 25, 33, 41\}$ | $0.4997 \pm 0.0031$ |

Table 8: Proxy Measure Sensitivity Analysis for $\mathcal{D}_{A2}$ (Score-Hessian Correlation). Default configuration is every 10 steps, Pearson correlation, 50 eigenvalues, all timesteps. Strided measurements tend to be robust and less noisy. Behaviours with respect to timestep ranges are expected: Gaussianity holds more strongly in earlier stages but weakens as the distribution sharpens into the complex, non-Gaussian data manifold.

| Test | Hyperparameter | $\mathcal{D}_{A2}$ | Test | Hyperparameter | $\mathcal{D}_{A2}$ |
|---|---|---|---|---|---|
| | every 1 | 0.708±0.172 | Correlation | Pearson | 0.981±0.011 |
| Timestep | every 2 | 0.923±0.042 | Method | Spearman | 0.988±0.021 |
| Interval | every 5 | 0.976±0.016 | | | |
| | every 10 | 0.994±0.006 | | 10 | 0.957±0.029 |
| | every 20 | 1.000±0.001 | | 20 | 0.974±0.005 |
| | all | 0.979 | Top-$k$ | | |
| Timestep | early $t \in [35, 49]$ | 0.995 | Eigenvalues | 50 | 0.980±0.010 |
| Range | mid $t \in [15, 35]$ | 0.982 | | 100 | 0.980±0.013 |
| | late $t \in [0, 15]$ | 0.893 | | 200 | 0.979±0.012 |
| | | | | 500 | 0.980±0.011 |

**(A2) Gaussian Local Structure.** Assuming local Gaussianity implies a direct proportionality between score-norm and curvature magnitude. We evaluate this via the Pearson correlation:

$$\mathcal{D}_{A2} = \mathrm{corr}\big( \|s_\theta(x, c)\|^2, -\mathrm{tr}(H_c(x)) \big).$$

This tests whether the identity in Lemma 4.1, originally derived for Gaussian densities, holds approximately for the learned distributions.

**(A3) Sharpness Persistence.** This assumption states that curvature patterns separating memorized and non-memorized samples persist along the reverse trajectory. We measure the temporal predictability of the curvature landscape through *temporal autocorrelation*. We first construct a spatio-temporal matrix $M \in \mathbb{R}^{T \times N}$ from the Hessian magnitudes. Each spatial feature (column) is then standardized to isolate its unique *evolution pattern* independent of absolute magnitude. We then compute the autocorrelation across several time lags $\tau$, aggregating scores by taking the median correlation across all positions for each lag to ensure robustness. A high final score, averaged across lags, indicates a stable and predictable evolution, upholding the assumption.

Table 9: Proxy Measure Sensitivity Analysis for $\mathcal{D}_{A3}$ (Sharpness Rank Persistence). Default configuration: Jaccard metric with lags $(1, 2, 4)$, all timesteps.

| Test | Hyperparam. | $\mathcal{D}_{A3}$ | Test | Hyperparam. | $\mathcal{D}_{A3}$ |
|---|---|---|---|---|---|
| | lag=1 | 0.324±0.012 | | lag_1 | 0.098±0.021 |
| Temporal | lag=2 | 0.320±0.012 | Jaccard: | lag_1_2 | 0.096±0.022 |
| Autocorr: | lag=3 | 0.316±0.013 | Lag Config | lag_1_2_4 | 0.095±0.022 |
| Max Lag | lag=5 | 0.306±0.014 | | lag_1_3_5 | 0.093±0.022 |
| | lag=10 | 0.268±0.016 | | lag_2_4_8 | 0.090±0.022 |

Alternatively, we examined *Hotspot Jaccard Persistence* by measuring the stability of the *locations* of the highest-curvature regions (eigenvalue "hotspots"). We identify the set of dimensions corresponding to the top-$q$ eigenvalues at different timesteps and compute their Jaccard similarity. High similarity indicates that the model's focus on specific geometric directions is stable. This is sensitive to the quantile hyperparameter $q$.

**(A4) Covariance Eigenspace Alignment.** Lemma 4.2 requires conditional and unconditional covariance matrices to commute, with aligned eigenspaces. We measure the average singular value of their eigenspace overlap:

$$\mathcal{D}_{A4} = \frac{1}{d} \sum_{i=1}^{d} \sigma_i\big(V_c^\top V_u\big),$$

where $V_c$ and $V_u$ are eigenvector matrices of $\Sigma_{t,c}$ and $\Sigma_t$, and $\sigma_i(\cdot)$ denote singular values. Higher $\mathcal{A}_{eig}$ indicates stronger commutativity.

**(A5) Mean-Field Gaussian Prior.** While we can trivially validate the implementation of the initial noise generator, a numerical diagnostic provides a more rigorous and quantitative verification of this property. The assumption $x_T \sim \mathcal{N}(0, I)$ underlies both early-step detection and the SAIL

Table 10: Proxy Measure Sensitivity Analysis for $\mathcal{D}_{A4}$ (Eigenspace Alignment). Default configuration: 20 samples, 1000 features, 50 eigenvalues.

| Test | Hyperparam | Mem | Non-Mem | Test | Hyperparam | Mem | Non-Mem |
|------|-----------|-----|---------|------|-----------|-----|---------|
| N Samples | 5 | 0.089±0.113 | 0.423±0.079 | N Features | 100 | 0.453±0.214 | 0.871±0.039 |
| | 10 | 0.107±0.116 | 0.433±0.079 | | 500 | 0.184±0.170 | 0.571±0.081 |
| | 15 | 0.111±0.117 | 0.414±0.079 | | 1000 | 0.113±0.122 | 0.414±0.086 |
| | 20 | 0.109±0.115 | 0.403±0.086 | | 2000 | 0.063±0.073 | 0.257±0.069 |
| | 30 | 0.114±0.126 | 0.414±0.085 | | 5000 | 0.028±0.035 | 0.124±0.043 |
| | 50 | 0.144±0.146 | 0.469±0.097 | | | | |

objective. We test adherence by comparing empirical initial latents against the Gaussian prior with a Kolmogorov–Smirnov (KS) test, yielding a $p$-value:

$$\mathcal{D}_{A5} = \text{KS}\big(\{x_T\}, \mathcal{N}(0, I)\big).$$

Table 11: A5 Sensitivity Analysis: Default option are underlined.

| Gaussianity Test | | Subsample Fraction | | Pooling | |
|------|------|------|------|------|------|
| Kolmogorov-Simonov | $0.8124 \pm 0.22$ | 5% | $0.7679 \pm 0.16$ | Overall | $0.8124 \pm 0.22$ |
| Shapiro-Wilk | $0.5695 \pm 0.23$ | 10% | $0.7473 \pm 0.19$ | Channels (avg) | $0.7846 \pm 0.21$ |
| Anderson-Darling | $0.7527 \pm 0.10$ | 25% | $0.7190 \pm 0.26$ | Blocks (avg) | $0.7861 \pm 0.19$ |
| Jarque-Bera | $0.4540 \pm 0.30$ | 50% | $0.8171 \pm 0.21$ | | |
| D'Agostino-P. | $0.4541 \pm 0.30$ | 100% | $0.8124 \pm 0.22$ | | |

**(A6) Boundary Regularity.** The integration-by-parts arguments used in Jeon et al. (2024) require $p(x)s(x) \to 0$ as $\|x\| \to \infty$ and finite $\mathbb{E}\|s(x)\|^2$. To detect violations, we compute a *score explosion indicator*:

$$\mathcal{D}_{A6} = \frac{\max_x \|s_\theta(x)\|}{\mathbb{E}_x[\|s_\theta(x)\|]}.$$

Values close to one indicate stable score magnitudes, while large ratios signal potential divergence, undermining the trace–score identity beyond Gaussian settings.

Together, these diagnostics allow us to empirically evaluate whether the assumptions underpinning Lemmas 4.1-4.3 hold in practice. As shown in Table 4, weaker adherence (e.g., in LaVie) correlates with degraded detection AUROC, confirming that deviation from these regularity assumptions contribute to reduced metric efficacy.

# D  EXTENDED CONTROLLED EXPERIMENTS

In Sec. 5, we presented results from controlled experiments targeting structured priors (D4). This appendix provides the full experimental details for all four design choices (D1-D4), illustrating how specific training protocols can systematically violate the geometric assumptions that memorization detection metrics rely upon.

**(D1) Non-Standard Objectives** To simulate the effect of auxiliary loss terms, we design an experiment that directly targets (A1) Unbiased Score Estimation. We introduce a penalty term to the standard diffusion loss that systematically biases the model's score predictions. Specifically, the loss function is modified to $\mathcal{L} = \mathcal{L}_{\text{LDM}} + \lambda \cdot \mathbb{E}[\text{ReLU}(\|\epsilon_{\text{pred}}\| - 0.7\|\epsilon_{\text{target}}\|)]$, which encourages the model to under-predict the magnitude of the noise. By training a series of models with varying penalty strengths $\lambda \in \{0.0, ..., 1.0\}$, we create a controlled degradation of the (A1) assumption. The results, shown in the top row of Fig. 6 (blue plots), confirm a strong correlation. As our diagnostic for (A1) Score Matching Consistency decreases (indicating a greater violation), the Hessian ROC performance consistently degrades, demonstrating the critical dependence of the metric on unbiased score estimates.

**(D2) Multi-Stage Distribution Shift** To model the fine-tuning process common in multi-modal systems, we implement a two-stage protocol where a baseline model trained on CIFAR-10 is subsequently fine-tuned on MNIST data (resized and converted to RGB). The degree of distribution S=shift is controlled by a parameter $\lambda \in [0, 1]$ that dictates the proportion of MNIST data in the

training mix. This setup is designed to induce violations in (A2) Gaussian Local Structure and (A4) Covariance Commutativity by warping the learned probability landscape. Unlike the other targeted interventions, this experiment did not yield a clear, monotonic relationship between the degree of distribution shift ($\lambda$) and the measured assumption violations or the final metric performance. Sharpness persistence (A3) and covariance commutativity (A4) are most fragile under partial shifts (60–80% MNIST), where mixed distributions misalign geometry, but they partly recover once training is dominated by the new modality. This suggests that while distribution shifts are a contributing factor to metric failure in real-world models, the interaction is complex; the model's adaptation to a new domain does not appear to be a simple linear interpolation, and the geometric consequences are less predictable, warranting further investigation.

**(D3) Non-monotonic Schedulers and Alternative Parameterization** To investigate the impact of the denoising path itself, we train models using `EulerDiscreteScheduler` as opposed to a baseline `DDPMScheduler`. This modification alters the dynamics of the reverse process, which is hypothesized to disrupt the temporal consistency of geometric features, thereby violating (A3) Sharpness Persistence and (A4) Covariance Commutativity. The results, shown in the bottom row of Fig. 6 (orange plots), reveal a dramatic effect. Models trained with the Euler scheduler exhibit a catastrophic drop in detection performance, which is strongly correlated with severe violations of several geometric assumptions, particularly (A2) Score-Hessian Proportionality and (A4) Eigenspace Alignment. The sharp, "step-function" like relationship in these plots indicates that certain schedulers can induce a regime change that fundamentally breaks the geometric conditions required for the detection metric to function.

**(D4) Structured Priors** This experiment, detailed in Sec. 5 of the main text, directly violates (A5) Mean-Field Gaussian Prior. We train and evaluate models using initial noise $x_T$ that has an imposed structure: either spatial (radial decay patterns) or frequency-based (low-pass filtered noise). The strength of this structure is controlled by a parameter $\lambda$. As shown in the main text, this direct violation of (A5) and its resulting downstream effects on other assumptions correlate strongly with a degradation in detection performance.

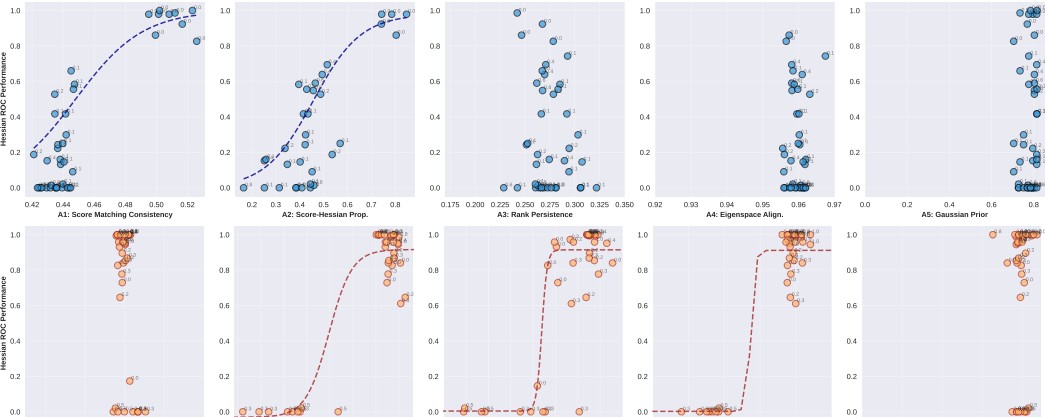

Figure 6: Detection performance (Hessian ROC) versus assumption adherence under controlled interventions. **Top Row (Blue):** Models trained with a **(D1) Non-Standard Objective** show a clear correlation between degraded (A1) consistency and lower ROC performance. **Bottom Row (Orange):** Models trained with a **(D3) Alternative Scheduler** exhibit a catastrophic drop in performance, linked to severe violations of (A2) and (A4). Each point represents a trained model.

## E  ADDITIONAL VISUALS

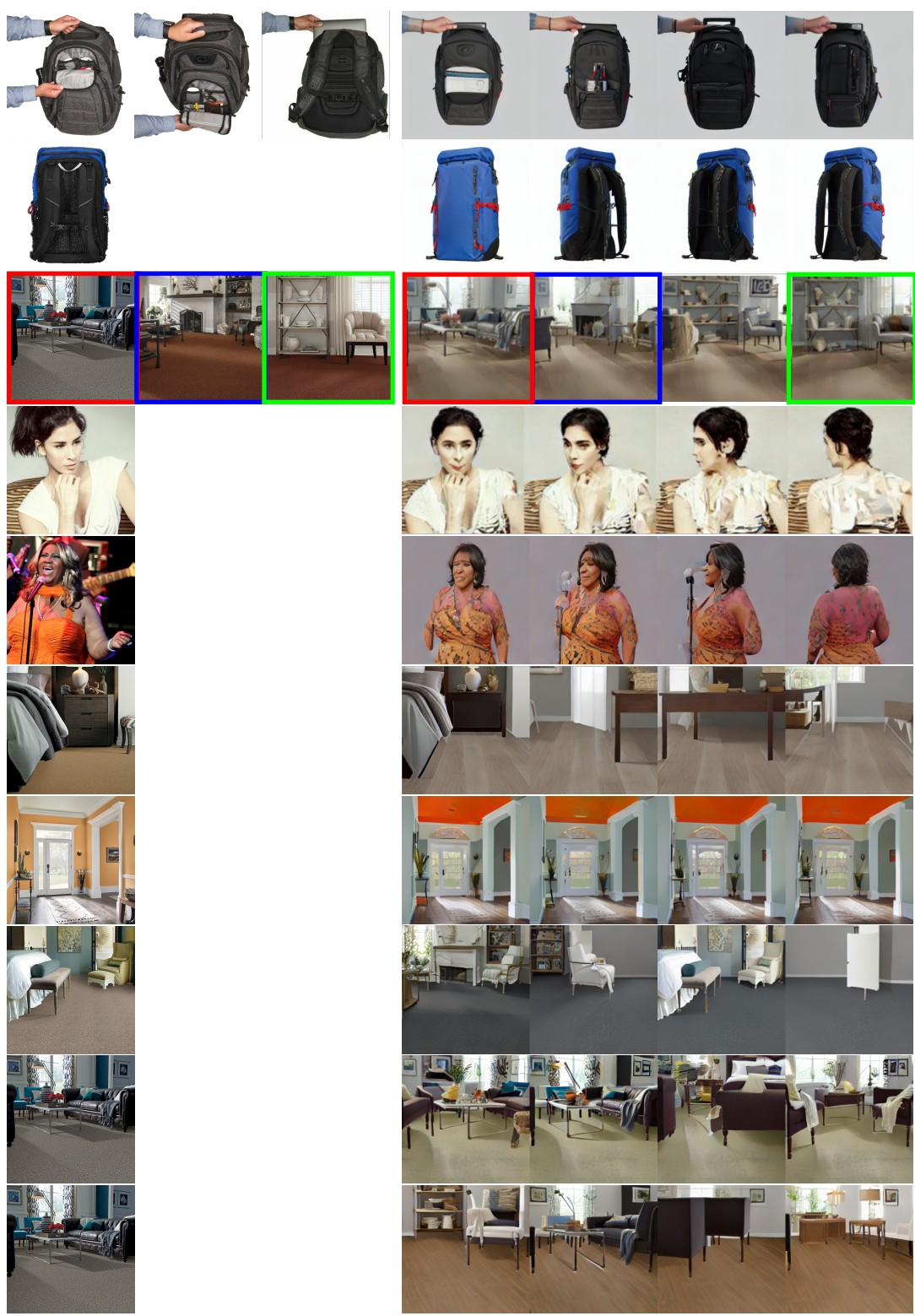

Figure 7: Examples of memorized samples inherited by MVDream from SD pre-training stage.

Figure 8: Examples of memorized samples generated by LaVie from both training stages.

Figure 9: Examples of memorized samples generated by DiffSplat from the Objaverse training stage.

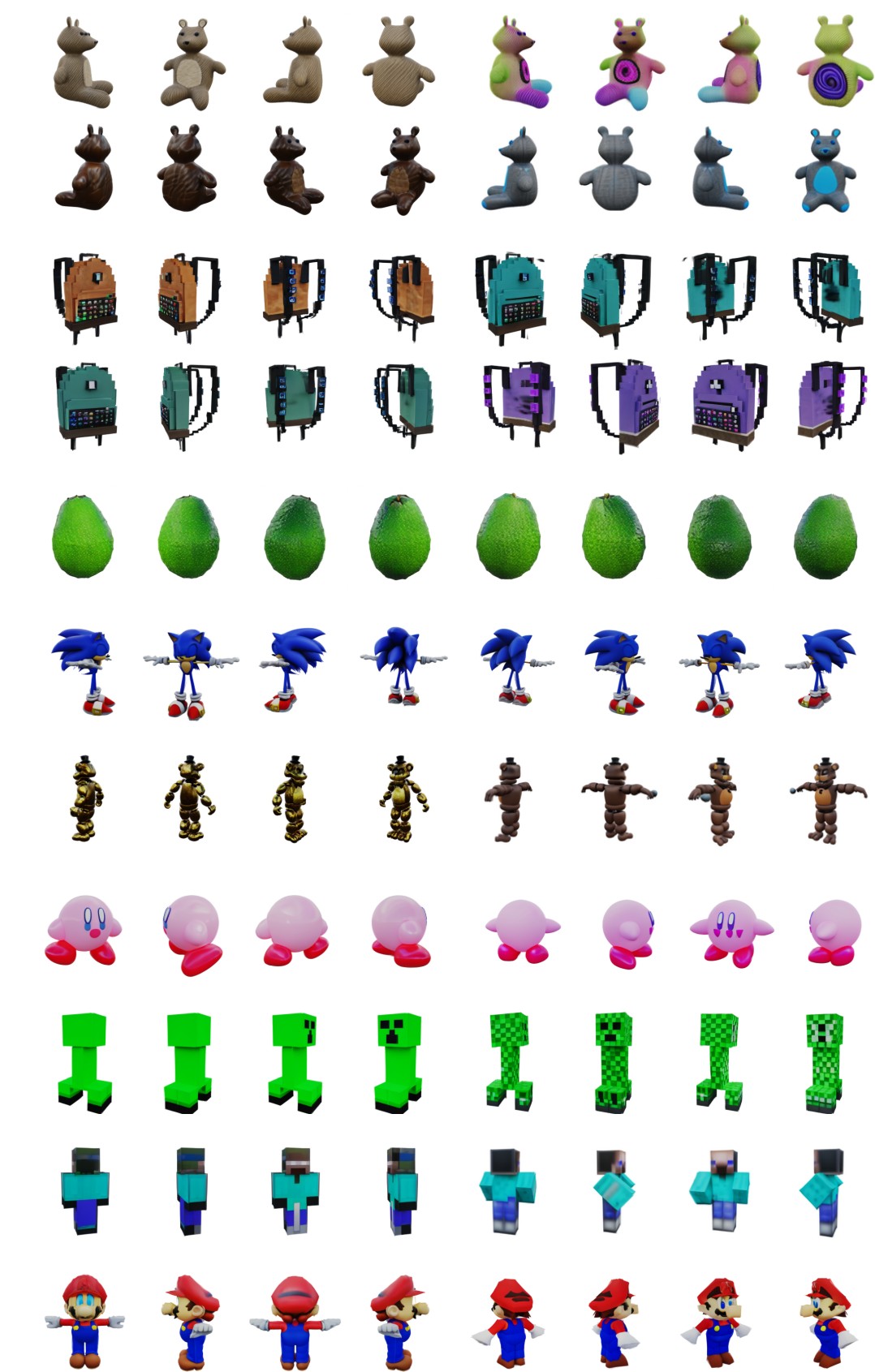

