# OpenReview forum: "On the Diminishing Reliability of Reference-Free Memorization Detection in Modern Diffusion Models"
_ICLR.cc/2026/Conference — ICLR 2026 Conference Withdrawn Submission_

### Official Review · Reviewer_mUf7 · 2025-10-18

**Soundness:** 2
**Presentation:** 3
**Contribution:** 1
**Rating:** 0
**Confidence:** 5

**Summary:**

This paper investigates why reference-free memorization detection methods, such as CFG- and attention-based metrics, lose reliability when applied to modern multi-stage and multi-modal diffusion models, revealing that violations of their underlying geometric assumptions lead to degraded performance and unreliable evaluation of unlearning methods.

**Strengths:**

1. The paper presents a clear and logically structured analysis that systematically connects empirical observations with theoretical explanations.

2. The experimental results are well-visualized, with comprehensive figures that effectively illustrate the degradation trends across models and metrics.

**Weaknesses:**

1. Flawed baseline and misimplementation of CAE
Regarding CAE (Cross-Attention Entropy) and SSCD, the entire experimental foundation seems incorrect. The baseline results for both CAE and SSCD are far below those reported in the original papers — both methods achieve over 0.99 AUROC on Stable Diffusion v1.4, yet this paper reports much lower values. In addition, the implementation of CAE is clearly wrong. The authors claim that “lower entropy indicates concentrated attention on a few tokens, suggesting memorization.” However, according to the original paper, only certain heads focus on the trigger token, while others focus on the beginning token. The competition between the trigger and the beginning token causes dispersed attention, i.e., higher entropy, when memorization occurs. Conversely, when there is no memorization, attention concentrates on the beginning token, leading to lower entropy. This fundamental misunderstanding makes the implementation invalid and casts doubt on the reliability of all experimental results. Since the entire paper’s conclusions rely on these experiments, I recommend a strong rejection.

2. Missing annotation in Appendix E
Appendix E lacks clear labeling, making it impossible to distinguish which images are training samples and which are generated samples. This seriously affects the interpretability of the visual results.

3. Poorly described experimental settings
The experimental settings are described with insufficient detail. For example, the paper does not specify how many images were used for SSCD, how many steps were taken, which specific CAE metric variant was used, or how many denoising steps were involved. These omissions make the experiments non-reproducible and the results unverifiable.

**Questions:**

As mentioned in Weakness 1 and Weakness 3, the experiments are unclear and terrible.

---

> ### Author Response · Authors · 2025-11-22
> **Response to Reviewer [mUf7]: Thank you for your thoughtful feedback!**
>
> We sincerely thank the reviewer for their thorough examination of our work and for voicing their concerns regarding metric implementation and baseline performance. We address each point below with clarification and proposed actionable revision plans.
>
> We appreciate that you find the paper presents a clear and logical structure, that you find the systematic connections between empirical observations and theoretical explanations valuable, and that you find the visualizations comprehensive and accessible.
>
> We believe that the concerns regarding reproducibility and implementation correctness have been sufficiently addressed in our discussions, with reference to the appendix and supplementary materials. The presentation of the method in the main text has also been extended through minor revisions. We would appreciate your reconsideration!
>
> ---
>
> **1. CAE Baseline Performance**
>
> We noticed the discrepancy between the reproduced results in Jeon et al. (ICML 2025) and the original results in Ren et al. (ECCV 2024). As noted by Jeon et al., this variance primarily stems from differences in non-memorized prompt set construction. Ren et al. used 500 GPT-4-generated prompts as negative samples (non-memorized prompt set). Following Hintersdorf et al. (2024), Wen et al. (2024), and Jeon et al. (2024), we employed a more heterogeneous non-memorized prompt set drawing from non-memorized prompts from LAION (Schuhmann et al., 2022), COCO (Lin et al., 2014), Lexica (2024), Tuxemon (HuggingFace, 2024), and GPT-4 (Achiam et al., 2023) generated samples.
>
> **Proposed revision:** We plan to extend our evaluation to include additional CAE variants ($E_{l=0…15}^{t=T}$ and $D$) and report AUROC as ranges across all CAE-E variants metrics and settings to demonstrate how metric efficacy varies with benchmark heterogeneity. Due to time constraints during rebuttal, we may focus on SD 1.4 baselines on naturally trained models first and extend to other settings subsequently.
>
> ---
>
> **2. SSCD Baseline Performance**
>
>
> We appreciate the opportunity to clarify the baseline context.
> The benchmark we utilized (Webster, 2023) was established in May 2023 and extended in various ways by Wen et al. (2024), Jeon et al. (2024), and Hintersdorf et al. (2024). The original SSCD (CVPR 2022) predates this benchmark, and the subsequent work applying SSCD to DM memorization evaluation (Somepalli et al., NeurIPS 2023) is contemporary to the original benchmark and predates the extended versions. To our knowledge, there are no official baseline AUROC metrics for SSCD in our exact evaluation setting.
>
> Additionally, our reported metric matches that of Wen et al. (2024), a recent reproduction.
>
> **We hope the metric consistency with reproduction in other SoTA works demonstrate the validity of our evaluation.**
>
> ---
>
> **3. CAE Metric Implementation and Polarity**
>
> We appreciate the reviewer’s careful attention to the nuances of cross-attention dynamics. We acknowledge that in the descriptive part of the text, we have misrepresented what was in fact implemented correctly in the code (supplementary materials).
>
> In our current presentation, our claim about entropy patterns is only valid in the regime where we conducted our evaluation ($t=T$ or $t \gg 0$) and only on certain layers, which is an oversimplification of the full picture presented in the original work and omits qualifications on attention head, timestep, and layer dependencies.
>
> We have expanded relevant sections (Sec 3 and Apx. B.2.3) to convey the dynamics throughout timesteps and between layers/heads discovered by the original work..
>
> The actual implementation follows the formulation and original implementation introduced by Ren et al. (ECCV 2024) and can be found in supplementary materials (evaluation/metrics/*.py), which correctly computes entropy across timesteps and layers.
>
> We would like to point out that CAE's (as with other metrics) core efficacy lies in **the separability between classes, not the polarity of the numerical values** (which flips across model layers at t=T, as shown in the original work as well as tables below, while **maintaining high separability in many layers**).
>
> Therefore, while we fully committed to correct Section 3.1.2 and in Apx. B.2.3 to include the important attention splitting behaviours as well as polarity reversal at different layers and timesteps, we would like to clarify that the implementation is correct and consistent with the original paper, and the reported results remain valid.

---

> ### Author Response · Authors · 2025-11-22
> **Response to Reviewer [mUf7]: Continued**
>
> Despite the nature of our work being a cautionary tale on metric generalizability, we would like to highlight the efficacy and robustness of CAE under a wide range settings, which also serves as a sanity check for the validity of our implementation and interpretation of the original work:
>
> (1) **CAE performs on par of SoTA metrics** across variants of the Webster benchmark on SD1.4 (see tables below), with expected and reasonable variations across varied levels of non-memorized prompt set heterogeneity.
>
> (2) **CAE performs exceptionally well as an evaluation tool** in the setting of several unlearning method that involves neuron ablation and attention manipulation.
>
> (3) Through mechanisms different from CFG-based metrics, CAE faces challenges in some case: especially in the presence of engagement of additional conditioning mechanisms (camera embedding, class-level prefix or post-fix tokens, etc.), which may have blurred its strong separability in the original setting. However, **CAE remains stable compared to CFG-based metrics (more robust and less performance drop than most CFG-based metrics)** due to its relative independence from learned geometry.
>
> ---
>
> **Proposed revision**:
>
> We have expanded relevant sections and appendices to more comprehensively discuss cross-attention dynamics across different timesteps, conveying fully and precisely the insightful observations from Ren et al.
>
> Additionally, we noticed that the optimal layer choice for $E_{t=T}$ differs across test settings.
>
> **Webster-Ren Variant**
>
> | Layer/Mode | Polarity | AUROC | TPR@0.1 |
> | :--- | :--- | :--- | :--- |
> | Layer 0 | + | 0.9328 | 0.8551 |
> | Layer 1 | + | 0.7643 | 0.4319 |
> | Layer 2 | + | 0.9027 | 0.7304 |
> | Layer 3 | + | 0.9958 | 0.9971 |
> | Layer 4 | - | 0.8869 | 0.6232 |
> | Layer 5 | - | 0.9866 | 0.9652 |
> | Layer 6 | - | 0.8743 | 0.7043 |
> | Layer 7 | - | 0.8685 | 0.6870 |
> | Layer 8 | - | 0.7950 | 0.6696 |
> | Layer 9 | - | 0.6695 | 0.4899 |
> | Layer 10 | + | 0.8372 | 0.5275 |
> | Layer 11 | - | 0.6442 | 0.4754 |
> | Layer 12 | - | 0.5351 | 0.2290 |
> | Layer 13 | + | 0.9419 | 0.8464 |
> | Layer 14 | + | 0.9979 | 0.9942 |
> | Layer 15 | + | 0.9123 | 0.7710 |
> | Average | | 0.8466 | 0.6873 |
>
> **Webster-Hintersdorf Variant**
>
> | Layer/Mode | Polarity | AUROC | TPR@0.1 |
> | :--- | :--- | :--- | :--- |
> | Layer 0 | + | 0.7782 | 0.3657 |
> | Layer 1 | + | 0.5594 | 0.1366 |
> | Layer 2 | + | 0.7677 | 0.3588 |
> | Layer 3 | + | 0.9640 | 0.9074 |
> | Layer 4 | - | 0.9215 | 0.7153 |
> | Layer 5 | - | 0.9761 | 0.9468 |
> | Layer 6 | - | 0.9052 | 0.7269 |
> | Layer 7 | - | 0.8975 | 0.7106 |
> | Layer 8 | - | 0.8050 | 0.6620 |
> | Layer 9 | - | 0.7462 | 0.6273 |
> | Layer 10 | + | 0.7415 | 0.3634 |
> | Layer 11 | - | 0.6842 | 0.5069 |
> | Layer 12 | - | 0.7284 | 0.3889 |
> | Layer 13 | + | 0.7666 | 0.3657 |
> | Layer 14 | + | 0.9616 | 0.9097 |
> | Layer 15 | + | 0.7862 | 0.5255 |
> | Average | | 0.8118 | 0.5761 |
>
> We propose to report the **best layer** ($l=5$, $t=T$) on Webster-Hintersdorf variant of the benchmark and amend this layer-choice ablation in the appendix.
>
> The polarity column uses + to indicate higher entropy → memorized, and - to indicate the reversed pattern.
>
> ---
>
> **4. Reproducibility**.
>
> We have provided all relevant hyperparameters in Appendix B.2.3 and Section 3.1.1 ($N=4$ for all experiments).
>
> Additionally, complete implementation code is included in the supplementary materials (evaluation/metrics/*.py).
>
> In our recent revision, sensitivity analysis for each assumption diagnostic test are also amended to Appendix C.
>
> We believe our implementation faithfully follows the original work, and we would welcome the reviewer's specific feedback on any aspects that may require correction.
>
> ---
>
> **5. Missing Labels in Appendix Figures**
>
> Thank you very much for catching this! We appreciate the attention to detail, and will clearly label training and generated images in the next revision!
>
> ---
>
> **Summary of Action Plan**
>
> To address the reviewer's concerns, we commit to:
> - Report the metric of the best layer choice of CAE-$E$ on each arm of the Webster benchmark, as well as on our Objaverse and WebVid benchmarks. Amend layer-wise ablation as a supplementary discussion in Apx. B.2.3.
> - Expanding our review of cross-attention dynamics across heads, layers, and timesteps in Section 3.1.2 and in Apx. B.2.3 to capture the full picture.
> - Add missing labels in the Appendix figures for better readability.
>
> We are grateful for this opportunity to improve the rigor and clarity of our work, and would appreciate your reconsideration!

---

> > ### Comment · Reviewer_mUf7 · 2025-11-22
> >
> > Thank you so much for your comprehensive response. Some of the concerns have been solved.
> >
> > But for CAE, please use the first metric in Eq. (6) of their paper, instead of the second metric in Eq. (7). If you only use one layer and t = T, this is actually an unfair setting. Because they have said that the second metric is for fast testing and the model only need to inference one step. The first metric has a consistent polarity. Please read carefully and use the best baseline to compare instead of the weak one.
> >
> > Overall, I think this will not influence the main conclusion, but I think this carelessness is not helpful for this work, which makes people doubt the correctness of other baselines.

---

> > > ### Author Response · Authors · 2025-11-24
> > > **Many thanks for your prompt reply!**
> > >
> > > Dear Reviewer,
> > >
> > > Thank you so much for your prompt response! We are glad that some of your concerns have been addressed.
> > >
> > > 1. As stated in our response to point 1, we are committed to including the CAE-$D$ variant (Eq. 6 from the original paper) and the continual expansion of our evaluation. While these experiments are still running, we can confirm the polarity alignment and consistency with the full metric as specified in the original work.
> > >
> > > 2. We completely agree that relying solely on a single layer provides an incomplete view of the efficacy of CAE-$E$. However, we believe CAE-$E$ has its distinct merits: (1) significantly reduced caching overhead and minimal runtime, and (2) particular utility as a diagnostic tool for unlearning methods that involve neuron ablation and attention manipulation, where highly localised analysis is really valuable. (We are considering including a vectorised variant of CAE-$E_{l=[0,15]}^{t=T}$ in addition to CAE-$D$ where optimal layer at $t=T$ is chosen based on the specific task).
> > >
> > > We acknowledge that a "fair" comparison involves multiple considerations (i.e. if considering caching overhead, space complexity, wall clock runtime, there are many ways in which one can argue fairness, or the lack thereof).
> > > Our intention has never been to establish a competition between metrics or to favour a particular one. Rather, we aim to view all metrics as complementary tools, each with unique strengths. Through comprehensive evaluation, our goal is to identify when and where we can maximize the power of each metric, while also highlighting scenarios where practitioners should exercise caution due to potential reliability limitations.
> > >
> > > We appreciate your vigilance in ensuring the rigor of our baseline comparisons, and we will ensure that CAE-$D$ is properly included and reported alongside CAE-$E$ and others, to provide a complete picture.

---

> > > > ### Comment · Reviewer_mUf7 · 2025-11-24
> > > >
> > > > Thank you for the follow-up. We appreciate the clarification and your continued engagement. We will wait for the additional experimental results before commenting further.

---

### Official Review · Reviewer_wwos · 2025-10-28

**Soundness:** 2
**Presentation:** 2
**Contribution:** 2
**Rating:** 4
**Confidence:** 3

**Summary:**

This paper systematically examines the reliability of reference-free memorization detection metrics for diffusion models in multi-stage/multimodal settings, and further evaluates whether these metrics can reliably verify dememorization success. It provides a geometric/statistical failure analysis with diagnostics and controlled interventions.

**Strengths:**

1.The paper systematically examines the applicability limits of T2I-era reference-free memorization detection metrics in video/3D/multi-stage pipelines, aligning with where diffusion models are rapidly expanding.

2.Evaluates both naturally trained models and multiple post-hoc dememorization methods across image, video, and 3D, improving external validity.

2.AUROC shrinkage and distribution overlap visualizations are intuitive.

**Weaknesses:**

1. While the paper diagnoses why reference-free metrics fail in video/3D settings, it does not translate its theoretical analysis into new or adapted metrics tailored to these modalities.

2. The compared models differ in resolution, sampling steps, CFG strength, scheduler, and latent space; without rigorous alignment and sensitivity analyses, the conclusions may be driven by implementation details rather than underlying phenomena.

**Questions:**

1.Have you evaluated the interaction between CFG and the number of sampling steps, and also performed  different CFG analyses?

2.Do you provide a systematic attribution of misclassified cases? Which categories most commonly cause reference-free metrics to fail?

---

> ### Author Response · Authors · 2025-11-24
> **Response to Reviewer [wwos]: Thank you for your constructive feedback!**
>
> We sincerely thank the reviewer for their thoughtful evaluation and for recognizing that our systematic examination of applicability limits aligns with where diffusion models are rapidly expanding, that our evaluation across naturally trained models and multiple mitigation methods improves external validity, and that our visualizations are intuitive.
>
>
> We appreciate the opportunity to clarify our experimental controls, address the questions raised, and discuss how we can strengthen the work. We would appreciate your reconsideration of the score following this discussion.
>
>
> ---
>
> **1. Contribution is Diagnostic Rather Than Methodological**.
>
> We understand that "Training-Aware or Universal Detection" is the ultimate goal. However, our observations demonstrate that developing these strategies is a nontrivial research undertaking that depends heavily on specific model architectures and training pipelines.
>
> We wish to not distract from the primary contribution of this paper, which is **the discovery and benchmarking of a previously undocumented limitation** that is currently overlooked in many existing works. By identifying this issue, hypothesizing its causes, and providing diagnostic tests, we establish a necessary foundation. We believe the complexity of a universal solution (or a suite of tailored-to-model solutions) warrants a separate, dedicated study.
>
> ---
>
> **2. Controlling for Model Parameter Heterogeneity**
>
> This is a very valid concern. We clarify that our evaluation strategy balances two goals:
>
> **(1) Real-world applicability:** Evaluating off-the-shelf models as they are deployed provides the most impactful assessment of metric reliability in practice. These models represent the actual use cases where practitioners need reliable detection, and their training and inference configuration are difficult to fully control.
>
> **(2) Controlled validation:** To supplement the real-world model evaluation where model parameter heterogeneity is indeed a concern, our controlled experiments (Section 5) systematically isolate specific factors under controlled conditions to validate causal relationships between training design choices and metric failure.
>
> **Please refer to the answers below to Q1 for revision proposals.**
>
>
> ---
>
>
> **Q1: Have you evaluated the interaction between CFG and the number of sampling steps, and performed different CFG analyses?**
>
> We agree with the reviewer that many prior works have shown that **training** time configurations, such as resolution, model capacity, and augmentation, impact the **prevalence** of memorization in the resultant model. However, how **inference** settings may impact (1) the **prevalence** of memorization and (2) the **efficacy of detection**, is less well understood, and could be a great addition to our evaluation.
>
> Several metrics’ original studies have explored how metric hyperparameter choices can impact efficacy. We propose to mirror these explorations and expand them beyond SD, prioritizing experiments on MVDream for the time being. We propose to add ablations on NDN with the following experiments:
>
> **(1) Resolution:** 512×512 (pre-training stage default resolution) vs 512×320 (tuning stage default resolution) on LaVie.
> **(2) Total sampling steps:** Varying total DDIM steps at inference time while computing the metric over the full trajectory.
> **(3) Temporal regime of evaluation:** Fixing total DDIM steps to default 50 and varying the range of steps over which we compute the metric.
> (4) CFG strength: We expect CFG primarily impacts diversity-based metrics (which trade diversity against prompt adherence).
> (5) Scheduler: We will test additional schedulers on real-world models, though, due to the model’s training setup, most applicable schedulers are monotonic.
>
> These experiments are currently underway, and we will update Appendix A upon completion.

---

> ### Author Response · Authors · 2025-11-24
> **Response to Reviewer [wwos]: Continued**
>
> **Q2: Do you provide a systematic attribution of misclassified cases? Which categories most commonly cause reference-free metrics to fail?**
>
> Our current analysis identifies failure modes through the lens of geometric diagnostics (Table 3), examining how training protocol design choices violate metric assumptions through the lens of our geometric framework.
>
> We interpret the reviewer's suggestion as recommending we complement this with an analysis of failure modes through the lens of **training data characteristics and high-risk sample commonalities.** This would provide a dual perspective on when and why metrics fail.
>
> **We have expanded Appendix A** to bridge these two complementary views, connecting geometric assumption violations to data characteristics that may have fostered increased misclassification. We observed that most metrics are more vulnerable when memorization and generalization behaviors co-exist within the same generation. For example:
> - Templated videos with temporal dilation/contraction, or minor variations in lighting, camera motion, or object motion.
> - 3D generations that memorise mesh geometry with texture variations.
> - Many-to-one memorization in multi-view generations that interpolate between multiple memorized images while maintaining cross-view consistency.
> - Cross-prompt memorization where regurgitation is triggered by a similar but distinct prompt (e.g. usually due to common prefix such as “Shaw Floor…” or templated prompts such as “A businessman…”)
>
> ---
>
> **Summary of Action Plan**
>
> - Expanding Appendix A.1 to systematically document both training and inference protocols, with ablation studies on sampling steps, temporal regimes, resolution, and other variations.
> - Expanding Appendix A.2 with failure mode analysis that examines training data characteristics and high-risk sample commonalities, providing various perspectives on when and why metrics fail.

---

### Official Review · Reviewer_2Ycj · 2025-11-01

**Soundness:** 2
**Presentation:** 2
**Contribution:** 2
**Rating:** 4
**Confidence:** 3

**Summary:**

This paper investigates the reliability of reference-free memorization detection metrics on complex, multi-modal diffusion models such as LaVie, MVDream, and DiffSplat. The authors find that these metrics degrade significantly under multi-stage or domain-shifted training, and analyze this through a geometric framework that links metric failure to violations of underlying theoretical assumptions (e.g., Gaussian locality, covariance alignment).

**Strengths:**

1. Timely and important topic. Memorization detection and model safety are high-interest issues in diffusion models.
2. Strong empirical scope. The experiments cover multiple metrics and models, providing a broad empirical overview.
3. The analysis connecting detection performance degradation to assumption violations is interesting.

**Weaknesses:**

1. **Low contribution.** The paper proposes no new detection metric, only diagnoses why existing ones fail. Since prior metrics were never designed for these complex training settings, observing degraded accuracy is somewhat trivial. While confirming this empirically and analyzing the cause is mildly interesting, a paper focused only on the limitations of existing methods without offering new methodology feels incomplete.
2. **Memorization bias is uncontrolled.** In Section 3, model-level memorization bias is a major confounding factor, but it is not considered. That is, the readers do not have information about how frequently a model reproduces training samples or how close generated outputs are to those samples. However, this is an important factor in the accuracy of detection methods. For instance, in SD 1.4, a prompt that reproduces memorized content 1/10 times versus 10/10 times would lead to very different detection accuracies, with the latter naturally yielding higher accuracy. Without quantifying or controlling this bias across different models, the evaluation lacks validity. Moreover, while diversity-based metrics such as Median SSCD and TL2 do not directly measure the proportion of generated but memorized images, they can implicitly indicate how strongly the model is biased toward a single image. The fact that MVDream, LaVie, and DiffSplat show low AUROC scores in these diversity metrics suggests that they are not strongly biased toward specific images — meaning their inherent memorization bias is low. Therefore, using such low-bias models to argue that detection metrics fail under complex training is misleading. A small-scale controlled experiment that systematically varies memorization bias under complex training settings would make this claim more convincing.
3. **Confounded experimental design.** Section 3 mixes modality and complex training, making it unclear which factor drives the performance drop. RL-fine-tuned models (e.g., GRPO, DPO) could also violate the assumptions in Section 4, but are not tested. Although Section 5 adds small-scale experiments, they are insufficient to support general conclusions about large-scale experiments.

To be acceptable, the work needs to go beyond diagnosing old metrics—it should either (a) propose a new detection method robust to complex training or (b) present a truly novel and concrete theoretical analysis. The contribution is too incremental now.

**Questions:**

1. Can authors conduct a controlled experiment to test how detection accuracy changes under different models with the same memorization bias?
2. Can authors isolate the effects of modality vs. complex training in large-scale models?

---

> ### Author Response · Authors · 2025-11-24
> **Response to Reviewer [2Ycj]: Thank you for your constructive feedback!**
>
> We sincerely thank the reviewer for their careful assessment and for raising important questions about experimental design and contribution scope. We appreciate that you find this a timely and important topic for the diffusion models community, that you recognize our strong empirical scope covering multiple metrics and models, and that you find the analysis connecting detection performance degradation to assumption violations interesting.
>
> We believe the concerns raised can be addressed through additional controlled experiments to isolate confounding factors and a clearer framing of our contribution as essential diagnostic groundwork for future method development. We appreciate your reconsideration of the score following this discussion.
>
> ---
>
> **1.  Contribution is Diagnostic Rather Than Methodological**
>
> We understand that "Training-Aware or Universal Detection" is the ultimate goal. However, our observations demonstrate that developing these strategies is a nontrivial research undertaking that depends heavily on specific model architectures and training pipelines.
>
> We wish to not distract from the primary contribution of this paper, which is **the discovery and benchmarking of a previously undocumented limitation** that is currently overlooked in many existing works. By identifying this issue, hypothesizing its causes, and providing diagnostic tests, we establish a necessary foundation. We believe the complexity of a universal solution (or a suite of tailored-to-model solutions) warrants a separate, dedicated study.
>
> ---
>
> **2. Controlling Memorization Bias**
>
> We appreciate the opportunity to further clarify both our evaluation design and expand our discussion on  various aspects of memorization bias and offer a more complete explanation for certain metric failure modes.
>
> Memorization bias here stands for regurgitation ratio per prompt across seeds. As Reviewer 2Ycj has pointed out, per-prompt memorization rate (regurgitation rate of 1/10 times versus 10/10 times) could vary across models, and could affect metric efficacy.
> This is indeed an important consideration, but its impact is more limited than it may initially appear due to the resolution at which we evaluate memorization. The following updates have been incorporated into our revision:
>
> - **We expanded Appendix A.1.3** to include a remark on memorization bias including (1) histograms (stratified memorization bias statistics) of SD1.4 and MVDream LAION subset (we will expand analogous analysis to other models and memorization subsets); (2) a brief discussion on other aspects of the “strength” of memorization.
> For inherited memorization from pre-trained datasets (e.g., LAION), there is a slight reduction in memorization bias but distributional trends remain consistent. The second-stage training appears to weaken, though not eliminate, the memorization tendencies established during pre-training. For newly acquired memorization from the second-stage dataset (e.g., Objaverse), memorization bias remains comparable to single-stage models. When a prompt triggers memorization, it typically does so consistently across seeds in both cases.
>
> - **Label Resolution**: Our memorization labels operate at the per-sample level (individual prompt-seed pairs, i.e., individual denoising trajectories), not per-prompt level. The majority of metrics we evaluate operate at per-sample resolution.
> These metrics receive a memorization label for each individual sample and predict memorization for that specific sample, making them inherently immune to regurgitation ratio bias.
> While the regurgitation ratio (how often a prompt memorizes across different seeds) impacts prompt level cross-seed metrics i.e. diversity metrics, other metrics evaluate each trajectory independently and this aspect is naturally controlled for.

---

> ### Author Response · Authors · 2025-11-24
> **Response to Reviewer [2Ycj]:  Continued**
>
> **3. Confounded Experimental Design - Modality vs. Training Complexity**
>
>
> Thank you for this important observation. We clarify that the key factor is **training design choices that violate metric assumptions** (neither modality nor sole complexity). Models with complex training protocols in the absence of assumption violations may still work well.
>
> Modality happens to be a very natural and common context in which these violations occur in diffusion models. We understand that the presentation could have caused confusion.
>
> Reviewer 2Ycj's suggestions are excellent examples for isolating training complexity effects in the image domain. We are eager to incorporate recent works (such as https://github.com/XueZeyue/DanceGRPO). However, there are several practical challenges which make it difficult to complete these experiments during the discussion period: (1) We need to retrain or reach original authors for SD checkpoints as only Flux checkpoints are available as of now. (2) Before committing to incorporating them, we will also need to evaluate the prevalence of memorization to ensure sufficient memorized samples are retained/inherited after fine-tuning to ensure statistical measurability.
>
> Our controlled experiments (Section 5) serves the purpose of isolating the effect of specific assumption violations within the same modality and model family. We recognize that extending similar controlled comparisons to large-scale "real world" capacity models would be valuable, though the substantial computational resources required would be better suited for a dedicated follow-up study.
>
> ---
>
>
> **Responses to Questions:**
>
> **Q1: Can authors conduct a controlled experiment to test how detection accuracy changes under different models with the same memorization bias?**
>
> We have expanded Appendix A to include memorization bias statistics of SD1.4 and MVDream LAION subset but plan to extend this to other models and data sources.
>
> Additionally, we wish to clarify that our memorization labels operate at the per-sample level (individual prompt-seed pairs, i.e., individual denoising trajectories), not at the per-prompt level.
> The regurgitation ratio (how often a prompt memorizes across different seeds) is therefore orthogonal to most metric evaluations except for cross-seed (diversity) metrics.
>
>
> ---
>
>
> **Q2: Can authors isolate the effects of modality vs. complex training in large-scale models?**
>
> We appreciate this suggestion. However, conducting large-scale controlled experiments that systematically vary training complexity while holding modality constant (or vice versa) would require training many full-scale models with carefully controlled variations, an extremely resource intensive process.
>
> We believe our current experimental design provides strong complementary evidence: Section 3 offers large-scale evaluations across real-world models, establishing the breadth and practical prevalence of the problem, while Section 5 provides tightly controlled, smaller-scale experiments that isolate specific design choice factors.
>
> ---
>
> **Summary of Action Plan**
>
> - We have expanded the memorization bias analysis to encompass three dimensions of memorization bias strength.
>
> - Conduct preliminary investigations and experiments on RL-finetuned models (e.g., DanceGRPO) or similar tuning protocols to further isolate training complexity effects within the image domain.
>
> We are grateful for this opportunity to improve the rigor and clarity of our work, and we would appreciate your reconsideration of the score in light of these clarifications and planned revisions.

---

> > ### Comment · Reviewer_2Ycj · 2025-11-28
> >
> > Thank you for the detailed response. However, before I reconsider my evaluation, I would like to clarify that the experiment corresponding to **W2. Controlling Memorization Bias** appears misaligned with what I originally intended. The rebuttal reports and interprets only the memorization bias histograms of Stable Diffusion 1.4 and MVDream-LAION. In contrast, what I requested was:
> >
> > *“Given a set of conditions (e.g., prompts) inducing similar memorization bias across different models, does applying the same mitigation method yield different detection AUROCs?”*
> >
> > I understand the authors’ claim that model-specific complex training procedures and domain differences may influence detection performance. However, the current experiment does not control for memorization bias across test samples. As shown in the histogram, MVDream-LAION exhibits much lower bias compared to Stable Diffusion 1.4, and detection methods are known to perform poorly when bias is low. Therefore, the fact that Stable Diffusion 1.4 (dotted gray line) achieves the highest AUROC in Figure 2 is expected given this imbalance in test sample bias.
> >
> > Accordingly, what I am asking is whether the same observation shown in Figure 2 persists even when the test samples are matched in memorization bias. Since the authors mentioned that other experimental extensions are difficult, I am willing to re-evaluate the paper once this specific experiment is conducted properly. Additionally, two minor issues:
> >
> > - In response to W2, the rebuttal states that the discussion was moved to A.3 in the Global Comments and to A.1.3 in the reviewer-specific section, but neither section exists—the content appears instead under A.1.2.
> > - I strongly recommend highlighting revised parts in a different color. It is currently very difficult to locate changes.
> >
> > I hope the authors provide a rebuttal that best enables the paper to be evaluated accurately and fairly.

---

### Official Review · Reviewer_64ic · 2025-11-01

**Soundness:** 2
**Presentation:** 3
**Contribution:** 3
**Rating:** 4
**Confidence:** 4

**Summary:**

This paper investigates the reliability of reference-free memorization detection metrics when applied to modern diffusion models (DMs) that go beyond standard text-to-image (T2I) training. While these metrics have shown promise for identifying memorization in Stable Diffusion, the authors show their performance degrades substantially in multi-modal or multi-stage models such as LaVie, MVDream, and DiffSplat. The study (1) systematically evaluates multiple CFG- and diversity-based metrics (NDN, HED, BE, SSCD, TL2, CAE) across diverse architectures, (2) examines their reliability in assessing post-training “unlearning” or memorization-mitigation methods, and (3) links the drop in reliability to violations of core geometric assumptions underlying these metrics. Empirical analysis and controlled experiments demonstrate that common training modifications can systematically invalidate these assumptions, explaining the degradation. The paper concludes by emphasizing the need for training-aware detection frameworks to ensure dependable safety evaluation of next-generation diffusion models.

**Strengths:**

1. The paper identifies a previously underexplored limitation: most memorization detection metrics are validated only on vanilla T2I diffusion models, yet are now routinely applied to much more complex systems (multi-modal, multi-stage, 3D/video).
2. The authors’ systematic evaluation and theoretical diagnosis represent a new perspective on the robustness and transferability of safety-critical metrics. The mapping between training-protocol design choices and theoretical assumption violations (Table 4) is an especially creative conceptual bridge.
3. The work carries high relevance for both safety and the evaluation of generative models. As diffusion architectures diversify, understanding when reference-free metrics fail is crucial for privacy auditing, model release, and regulatory compliance.
4. The paper is well written and easy to follow. Figures are informative and visually consistent.

**Weaknesses:**

1. The current experiments are insufficient. It is suggested to include more baseline methods in the comparative results to make the claim of “As diffusion models expand beyond the familiar text-to-image paradigm to encompass multi-modal and multi-stage training for 3D and video synthesis, the reliability of existing detection methods in these novel domains remains unclear” more convincing. Currently, only 3 methods (LaVie, MVDream, and DiffSplat) are being experimented with in total for these two additional modalities. For example, a recent work [1] that focuses on memorization in the video diffusion models has conducted experiments on ModelScope, VideoCrafter, and RaMViD, in addition to LaVie, and validated that they are all subject to the memorization problem. Thus, it would be interesting to see if the metrics tested in this paper also underperform on these additional video diffusion models, which could potentially strengthen the claim. [1] Investigating Memorization in Video Diffusion Models. arXiv, 2025.
2. Some diagnostics (Table 2) rely on proxy estimators that can be noisy or implementation-dependent. The authors acknowledge this, but could better quantify uncertainty or validate proxies on controlled synthetic models.
3. The analysis in Sec. 3.3 tests twelve mitigation techniques, but this section lacks clear grouping or interpretation of which categories fail and why. A summarized taxonomy (gradient-based vs. attention-based vs. concept-ablation) and correlation with metric families would strengthen Section 3.3.
4. While the paper calls for “training-aware detection strategies,” it does not sketch concrete formulations or early prototypes. Including one proof-of-concept adjustment could make the contribution more forward-looking.
5. The paper fails to completely follow the ICLR template, as the abstract is over-wide.

**Questions:**

Please check out the strengths and weaknesses sections.

---

> ### Author Response · Authors · 2025-11-22
> **Response to Reviewer [64ic]: Thank you for your constructive feedback!**
>
> We sincerely thank the reviewer for their thorough and constructive evaluation of our work.
>
> We are encouraged that you find this previously underexplored limitation valuable, that you recognize our systematic evaluation and theoretical diagnosis as offering a new perspective on robustness and transferability, and that you appreciate the high relevance of this work for both safety and evaluation of generative models.
>
> We appreciate the opportunity to strengthen our work by clarifying the evaluation scope choice, addressing questions about the generalizability of the observed phenomena, and including categorization of mitigation methods in the main text.
> We would appreciate your reconsideration of the score following the discussion!
>
> ----
> **1. Choice of evaluation scope among naturally trained diffusion models**
>
> We appreciate the opportunity to clarify our model selection criteria and address the broader question of generalizability.
>
> **Statistical necessity:** Evaluating detection metric performance requires sufficient ground-truth memorized samples for distributional analysis (ROC curves, precision-recall, statistical significance). Models like VideoCrafter (SD2.1-based) and ModelScope implement known memorization-reduction strategies: VideoCrafter inherits SD2.1 which underwent dataset curation, while ModelScope authors explicitly document deduplication preprocessing. Prior work demonstrates that such interventions - particularly deduplication and safety filtering - substantially reduce memorization prevalence.
> In preliminary screening, VideoCrafter and ModelScope yielded very few candidate memorized samples compared to LaVie's under identical detection protocols. This sparsity creates fundamental evaluation challenges: (1) insufficient statistical power for reliable metric comparison, and (2) near-degenerate precision-recall curves where class imbalance dominates signal. Our model selection prioritizes evaluability: we require sufficient memorization samples to meaningfully test whether metrics fail, succeed, or provide unreliable signals.
>
> **Testing falsifiable hypotheses:** Our selected models (LaVie, MVDream, DiffSplat) enable both confirmation and falsification of our central hypothesis. They exhibit diverse training protocols (multi-stage fine-tuning with distribution shifts, joint modality objectives, and domain-specific regularization) - allowing us to test whether specific architectural choices correlate with metric degradation. Importantly, if metrics had performed consistently well across these models despite architectural and training protocol diversity, that would have falsified our hypothesis about geometric assumption violations.
> RaMViD presents a complementary challenge: its disjoint training data (different from LAION/WebVid/Objaverse) would require constructing a new arm in our high-risk prompt benchmark with domain-appropriate prompt sets and ground-truth verification. While valuable for assessing cross-domain generalization, this represents orthogonal work requiring substantial additional resources. We acknowledge this as important future work.
>
> ----
> **2. Proxy Estimators and Uncertainty Quantification**
>
> As we acknowledged in the original submission, the noisiness of proxy measurements in Table 2 is a fundamental challenge when diagnosing geometric properties of high-dimensional diffusion models. We report mean ± standard deviation in Table 3 across 5 samples per configuration to characterize variability
>
> **Proposed Revision:** We acknowledge that proxy measurements in high-dimensional spaces carry inherent uncertainty. We have now expanded Appendix C.2 with Sensitivity analysis examining how key hyperparameters and implementation variations affect each diagnostic test.
> We encourage future metric users to refer to our analysis when selecting stable default hyperparameters. This also serves as a template for model developers who may need to re-test these parameters on future models with different geometric regularity conditions.
>
> Overall, the results confirm that the core observations regarding geometric violations reported in the main text remain valid and robust across different configurations.
>
> ----
> **3. Organization and Interpretation of Section 3.3**
>
> Thank you for highlighting this! Actually, we did include method categorization in Table 5 (Appendix B.3.1), but we recognize that it is not very visible nor explicitly referenced by the main text.
>
> **Proposed Revision:** We have made the category more prominent in the main text by incorporating it directly into Section 3.3 with visual distinctions (i.e. colour-coded method acronyms on radial plots along with explanation and a pointer to Table 5, Appendix B.3.1).

---

> ### Author Response · Authors · 2025-11-22
> **Response to Reviewer [64ic]: Continued**
>
> **4.  Contribution is Diagnostic Rather Than Methodological**
>
> We understand that "Training-Aware or Universal Detection" is the ultimate goal. However, our observations demonstrate that developing these strategies is a nontrivial research undertaking that depends heavily on specific model architectures and training pipelines.
> We wish to not distract from the primary contribution of this paper, which is the discovery and benchmarking of a previously undocumented limitation that is currently overlooked in many existing works. By identifying this issue, hypothesizing its causes, and providing diagnostic tests, we establish a necessary foundation. We believe the complexity of a universal solution warrants a separate, dedicated study.
>
> ---
>
> **Summary of Action Plan**.
>
> We commit to:
> - Discuss model selection rationale and representativeness in Appendix A.1.
> - Expand Appendix C.2 with sensitivity analysis for proxy measurements.
> - Restyle Section 3.3 plots by adding explicit colour coding for categorization for better organization, mirroring Table 5 (Appendix B.3.1).
> - Correct the abstract formatting to fully comply with the ICLR template.
>
> We are grateful for this opportunity to improve the rigor and clarity of our work, and we would appreciate your reconsideration of the score in light of these clarifications and planned revisions.

---

### Author Response · Authors · 2025-11-24
**List of Revision Items**

Dear all,

We would like to thank the reviewers for their valuable and constructive feedback.

We are grateful for your positive remarks regarding the significance of this topic and the novel connection established between memorization detection efficacy and distributional geometry regularity.

We will continue to update the draft as more additional ablations and experiments are done.

In the meantime we would appreciate your review, additional comments or feedback!

- Appendix A.1: Beyond the training protocol summary, we have now included inference-time hyperparameters for each model.
- Appendix A.3: We have expanded the memorization bias analysis to encompass three dimensions of memorization bias strength.
- Appendix C.2: We have added a sensitivity analysis for proxy measure hyperparameters.
- Section 3: We have updated the CAE-$E$ baseline and in the process of adding layer-wise ablation tables in Appendix B.2.3 for comprehensive documentation.
- Section 3: We have applied color coding to the unlearning method radial plots to align with the categorization in Table 5.
- Abstract: We have corrected the abstract margin formatting.


Upcoming additions:
- In line with our contribution, we are committed to the ongoing expansion of our evaluation framework. We will add the CAE-$D$ variant and incorporate more recently published metrics (InvMM, pLaplace). To better illustrate their relationship with other CFG-discrepancy based metrics, we have renamed the initial group to score-based metrics. The radial plots will be updated accordingly.
- Appendix A: We will include ablation studies for inference time parameters (e.g. resolution/frame/CFG scale/CFG time step subsets).
- Appendix E: We will expand the visuals and improve labelling.

---

### Note · Authors · 2026-01-15

I have read and agree with the venue's withdrawal policy on behalf of myself and my co-authors.